# iGRPO: Self-Feedback-Driven LLM Reasoning

## Abstract

Large Language Models (LLMs) have demonstrated considerable promise in solving complex mathematical problems, yet they still fall short of producing fully accurate and consistent solutions. Reinforcement Learning (RL) has emerged as a powerful framework for aligning these models with task-specific rewards, thereby improving their overall quality and reliability. In particular, Group Relative Policy Optimization (GRPO) has emerged as an efficient, value-function-free alternative to Proximal Policy Optimization (PPO). In this paper, we introduce Iterative GRPO (iGRPO), a two-stage extension of GRPO that boosts performance by incorporating a self-feedback mechanism. Specifically, iGRPO first samples multiple candidate responses (self-feedback stage) and selects the highest-reward completion as feedback for the second stage (self-reflection stage). By conditioning the model on this top-performing draft, iGRPO effectively guides the policy to generate refined solutions surpassing the best first-stage candidates. Through systematic comparisons with GRPO across different base models (e.g., Nemotron-H-8B-Base-8K, DeepSeek-R1 Distilled), we extensively validate the effectiveness of iGRPO. In addition, further training the OpenReasoning-Nemotron-7B model with iGRPO algorithm results in new state-of-the-art benchmarks of 85.62% and 79.64% on AIME24 and AIME25, respectively. These results underscores the potential of iterative, self-feedbackdriven RL approaches for advancing the reasoning performance.

## 1 Introduction

Reinforcement Learning (RL) has proven to be successful in improving reasoning capabilities of LLMs by optimizing against task-specific reward signals. Early successes in this direction include RL from Human Feedback (RLHF) for aligning LLMs with human intent, most notably in InstructGPT (Ouyang et al., 2022) and ChatGPT (Achiam et al., 2023), which have demonstrated that incorporating preference-based rewards can dramatically improve both the usability and correctness of model outputs. Recently DeepSeek-R1 (Guo et al., 2025) proposed a distinguishing feature which is the so-called zero configuration, wherein the RL process directly enhances the base language model. This breakthrough started several efforts which were targeted at replicating DeepSeek-R1s methodology or refining its underlying RL mechanisms (Zeng et al., 2025; Yu et al., 2025; Liu et al., 2025; Cui et al., 2025; Hu et al., 2025).

Yet, in the realm of complex reasoning, RL algorithms typically do not incorporate any form of *feedback or reflection* on the models own outputs. Humans, by contrast, rarely solve nontrivial problems in a single pass: they often iterate on initial drafts, identify mistakes, and refine their solutions based on internal feedback (Flower & Hayes, 1981; Simon, 2012; Braidotti, 2019; Flavell, 1979; Schön, 2017; Polya, 2014). There is growing evidence that self-feedback mechanisms can bolster multi-step reasoning and the capacity to correct errors (Madaan et al., 2023; Shinn et al., 2023). However, existing RL frameworks do not capitalize on this iterative refinement process, leaving a critical gap between how humans naturally solve problems and how LLMs are typically trained to do so.

Figure 1: **Iterative GRPO (iGRPO):** In the *Stage1* we draw multiple candidate completions and compute their relative rewards. We then select the highest-scoring completion which serves as *Self-Feedback*. In *Stage2*, we feed the *Self-Feedback* back to the model along with the original prompt and refine the outputs by generating another set of candidate completions.

In this work, we propose to fill this gap with *Iterative GRPO (iGRPO)* which is a powerful extension of GRPO (Shao et al., 2024). As illustrated in Fig. 1, our method operates in two stages. First, we draw multiple candidate completions from the model and compute their relative rewards via the group-based mechanism of GRPO. We then select the highest-scoring draft and this serves as the "first-draft" output of the model. We consider this highest-scoring response as a guide to improve the final output. Hence, it is provided as a self-feedback to the model. We feed it back to the model alongside the original prompt. By conditioning on this exemplar, the second stage encourages the model to refine and surpass its own best prior attempt. Notably, this design preserves the efficiency of GRPO while introducing only minimal extra overhead, as iGRPO still relies on the same set of group-based reward signals. In doing so, iGRPO offers a promising avenue for self-guided improvement, enabling LLMs to iteratively improve their reasoning capabilities.

We conduct a series of controlled experiments to compare iGRPO and GRPO under identical training conditions, using different base models trained on the Mathematics Aptitude Test of Heuristics (MATH) (Hendrycks et al., 2021) dataset. Specifically, we evaluate DeepSeek-R1 Distilled (Guo et al., 2025) and OpenMath-Nemotron (Moshkov et al., 2025) on an extensive array of mathematical reasoning benchmarks, including AIME24 (AI-MO, 2024a), AIME25 (OpenCompass, 2025), MATH500 (Lightman et al., 2023), AMC23 (AI-MO, 2024b), GSM8K (Cobbe et al., 2021), and Minerva Math (Lewkowycz et al., 2022). For models with 7B and 14B parameters, iGRPO consistently outperforms standard GRPO. Moreover, by leveraging iGRPO algorithm with OpenReasoning-Nemotron-7B model (NVIDIA, 2025) on the large-scale **AceReasonMath** (Chen et al., 2025b) dataset (R1, 2024), we push the state of the art on AIME24 and AIME25 to 85.62% and 79.64%, respectively. These findings underscore the effectiveness of incorporating a self-feedback stage into group-based RL optimization, particularly for complex mathematical reasoning tasks.

## 2 RELATED WORK

**RL Reasoning.** RL has become essential for refining large language models in logical and analytical tasks (Lambert et al., 2024). Early self-enhancement efforts such as STaR exploit verified responses and outcome-based sampling to iteratively refine chain-of-thought explanations. The o1 system (Jaech et al., 2024) then scaled this approach to record-breaking levels, with open-weight models such as DeepSeek-R1 (Guo et al., 2025) sometimes exceeding o1s performance. Furthermore, RL on procedurally generated puzzles (Xie et al., 2025) and even a single human demonstration (Wang et al., 2025) underscore RLs versatility in boosting mathematical reasoning. large-scale human-provided data (Wang et al., 2025).

**LLM Self-Learning.** LLM self-learning has recently emerged as a strategy for large language models to refine their reasoning and generation by leveraging feedback from themselves. SPIN and Self-Rewarding Language Models (Chen et al., 2024; Yuan et al., 2024) embody this approach by using the same model instance as its own reward model, boosting both discriminative and generative faculties. Other lines of

work employ self-play or verifier-based mechanisms for alignment (Kirchner et al., 2024; Ye et al., 2024), although unreliable rewards can hamper complex reasoning (Lambert et al., 2024). SPC (Chen et al., 2025a) and SPAG (Cheng et al., 2024) expand self-play with curated tasks and adversarial taboo scenarios, respectively, to hone critique capabilities. While these methods empower a model to generate and critique its own outputs, they often entangle the roles of reward provider and reward recipient within a single agent or paired agents. By contrast, our *iGRPO* framework focuses on using externally evaluated best-prior completions as an in-context guide, maintaining a clearer separation between the models generation process and its reward source. This design avoids problems related to self-confirmation loops and unreliable intrinsic rewards, while retaining the core self-improvement principle central to self-learning approaches.

**GRPO Improvements.** Recently, there have been efforts to refine and extend GRPO for large-scale LLM training. Dr.GRPO (Liu et al., 2025) highlights an overlooked source of bias in GRPOs token-level objectives and proposes removing the division by sequence length and group-level standard deviation to achieve an unbiased policy gradient. DAPO (Yu et al., 2025) addresses long-CoT (Chain-of-Thought) scenarios through dynamic sampling, decoupled clipping, and overlong reward shaping, aiming to reduce training instability and reward noise. GSPO (Zheng et al., 2025) takes a different perspective by operating at the sequence level, redefining importance ratios and applying sequence-level clipping to improve stability, particularly for Mixture-of-Experts models. While these approaches concentrate on stabilizing and refining the underlying optimization process, iGRPO moves in an orthogonal direction by introducing a two-stage mechanism that leverages externally evaluated completions to guide policy refinement.

## 3 METHODOLOGY

### 3.1 BACKGROUND: GROUP RELATIVE POLICY OPTIMIZATION

GRPO is a value-function-free variant of proximal policy optimization that leverages *group-based* relative rewards for advantage estimation. Let $\pi_\theta$ denote the current policy and $\pi_{\theta_{\text{old}}}$ the old policy. We begin with a pretrained language model $\mathcal{M}$ and a set of training instances $\mathcal{B} = \{(q, a)\}$, where $q$ is a prompt (e.g., a math problem) and $a$ is a reference answer. Given a prompt $q$, GRPO samples a group of $G$ candidate outputs $\{o_1, o_2, \ldots, o_G\}$ from $\pi_{\theta_{\text{old}}}$:

$$o_i \sim \pi_{\theta_{\text{old}}}(\cdot \mid q) \quad \text{for } i = 1, \ldots, G.$$

A reward model then evaluates each sampled output $o_i$, resulting in scores $\{r_1, \ldots, r_G\}$. GRPO normalizes these scores within the group to compute the advantage $\hat{A}_{i,t}$ at each token index $t$ of $o_i$:

$$\hat{A}_{i,t} = \frac{r_i - \text{mean}(\{r_1, \ldots, r_G\})}{\text{std}(\{r_1, \ldots, r_G\})},$$

where $\text{mean}(\cdot)$ and $\text{std}(\cdot)$ denote the sample mean and standard deviation of the groups reward scores. Note that all tokens $t$ in $o_i$ share the same advantage $\hat{A}_{i,t} = \hat{A}_i$, reflecting a single scalar reward for each sampled completion.

GRPO then updates the current policy $\pi_\theta$ by maximizing a clipped surrogate objective. Let

$$r_{i,t}(\theta) = \frac{\pi_\theta(o_{i,t} \mid q, o_{i,<t})}{\pi_{\theta_{\text{old}}}(o_{i,t} \mid q, o_{i,<t})}.$$

Then the GRPO objective is:

$$\mathcal{J}_{\text{GRPO}}(\theta) = \mathbb{E}\Big[q \sim P(Q), \{o_i\}_{i=1}^G \sim \pi_{\theta_{\text{old}}}\Big]$$

$$\times \frac{1}{G} \sum_{i=1}^G \frac{1}{|o_i|} \sum_{t=1}^{|o_i|} \Big[\min\Big(r_{i,t}(\theta)\,\hat{A}_{i,t},\ \text{clip}\big(r_{i,t}(\theta),\ 1-\epsilon,\ 1+\epsilon\big)\,\hat{A}_{i,t}\Big) - \beta\,D_{\text{KL}}\big(\pi_\theta \,\|\, \pi_{\text{ref}}\big)\Big], \quad (1)$$

where $\epsilon$ is the PPO clipping parameter, $\beta$ is a regularization coefficient on the KL divergence to a reference policy $\pi_{\text{ref}}$, and $|o_i|$ denotes the token length of completion $o_i$. The KL term $D_{\text{KL}}(\pi_\theta \| \pi_{\text{ref}})$ is approximated via an unbiased estimator Schulman (2020):

$$D_{\text{KL}}\big[\pi_\theta \,\big\|\, \pi_{\text{ref}}\big] \;=\; \frac{\pi_{\text{ref}}(o_{i,t} \mid q, o_{i,<t})}{\pi_\theta(o_{i,t} \mid q, o_{i,<t})} \;-\; \log \frac{\pi_{\text{ref}}(o_{i,t} \mid q, o_{i,<t})}{\pi_\theta(o_{i,t} \mid q, o_{i,<t})} \;-\; 1,$$

which remains guaranteed to be non-negative. By directly computing group-based advantages instead of estimating value functions, GRPO avoids the overhead of training a separate critic model, making it particularly appealing for large-scale language model fine-tuning.

## 3.2 ITERATIVE GRPO

We now introduce our two-stage iGRPO algorithm, which supplements the GRPO framework with a self-feedback mechanism. The key insight is to expose the policy model to its own best-scoring draft as a guiding example, thereby enabling *self-reflection* and further refinement of the generated output. Below, we first outline the conceptual steps of iGRPO and then present its mathematical formulation.

**Conceptual Overview.** For each prompt $q$, iGRPO proceeds in two main stages:

1. **Stage 1 (Self-Feedback).** Sample $N$ candidate completions $\{r_1, r_2, \ldots, r_N\}$ from the policy $\pi_{\theta_{\text{old}}}$, and evaluate each candidate with a reward model $R_\phi$. Identify the highest-scoring completion, denoted by $\hat{r}$, and treat it as a form of "feedback text."

2. **Stage 2 (Self-Reflection).** Form an augmented prompt

$$q' \;=\; \text{Concat}\big(q, \texttt{"<SelfFeedback>"}, \hat{r}\big),$$

   and sample another group of $G$ completions

$$\{o_1, \ldots, o_G\} \;\sim\; \pi_{\theta_{\text{old}}}(\cdot \mid q').$$

   Using these $G$ completions, compute group-based advantages as in GRPO and update the policy $\pi_\theta$ to reinforce high-reward outputs. Here "`<SelfFeedback>`" is a fixed textual delimiter.

Intuitively, this procedure exposes the model to an example of its own best prior attempt, encouraging improved solutions in the second generation phase. In Algorithm 1, we demonstrate the working principles of iGRPO. In the following, we describe how these two stages integrate into a single learning objective.

**Reward model.** Here $R_\phi$ is a rule based reward function. It extracts the final answer from each completion, returns 1 for a normalized exact match and 0 otherwise, and this scalar reward is used by GRPO and iGRPO for group normalization, advantage computation, and for choosing the best Stage 1 draft.

**Mathematical Formulation.** Let $\pi_\theta$ be the *current* policy, initialized from some $\pi_{\theta_{\text{old}}}$ at each training iteration, and $R_\phi$ be the reward model. For each prompt $q$, Stage 1 samples $N$ completions from $\pi_{\theta_{\text{old}}}$:

$$r_i \;\sim\; \pi_{\theta_{\text{old}}}(\cdot \mid q), \quad i = 1, \ldots, N.$$

We obtain scalar rewards $\{R_\phi(r_1), \ldots, R_\phi(r_N)\}$ and pick the highest-scoring completion:

$$\hat{r} \;=\; \underset{r_1, \ldots, r_N}{\arg\max} \; R_\phi(r_i). \tag{2}$$

We then augment $q$ with $\hat{r}$ to form $q'$, which is used in Stage 2:

$$q' \;=\; \big[q; \texttt{"<SelfFeedback>"}; \hat{r}\big].$$

---

**Algorithm 1** Iterative Group Relative Policy Optimization (iGRPO)

---

**Require: Inputs:** Pretrained policy $\pi_{\theta_{\text{init}}}$, reward model $R_\phi$, dataset $\mathcal{D}$, group size $G$, draft count $N$, clipping parameter $\epsilon$, KL coefficient $\beta$, number of iterations $I$, batch size $M$.

1: $\pi_\theta \leftarrow \pi_{\theta_{\text{init}}}$
2: **for** iteration $= 1, \ldots, I$ **do**
3:     $\pi_{\theta_{\text{old}}} \leftarrow \pi_\theta$
4:     Sample $M$ prompts $\{q^{(1)}, q^{(2)}, \ldots, q^{(M)}\}$ from $\mathcal{D}$
5:     **for** $k = 1, \ldots, M$ **do**
6:         **Stage 1 (SelfFeedback):**
        $\{r_i\}_{i=1}^N \sim \pi_{\theta_{\text{old}}}(\cdot \mid q^{(k)})$; select top draft $\hat{r}$ by reward $R_\phi$.
7:         **Stage 2 (Self-Refinement):**
        Augment prompt
$$q' = \big[q^{(k)}; \langle \text{SelfFeedback} \rangle; \hat{r}\big].$$
        $\{o_i\}_{i=1}^G \sim \pi_{\theta_{\text{old}}}(\cdot \mid q')$; compute rewards $\{r_i\}_{i=1}^G$.
        Normalize rewards to get advantages $\{\widetilde{r}_i\}$; apply GRPO update with clipped ratio.
8:         $\theta \leftarrow \theta + \eta \nabla_\theta \mathcal{J}_{\text{iGRPO}}(\theta)$                    ▷ gradient ascent
9:     **end for**
10: **end for**
11: **Return** $\pi_\theta$

---

Next, we draw a *group* of $G$ completions $\{o_1, \ldots, o_G\}$ from $\pi_{\theta_{\text{old}}}(\cdot \mid q')$. Each completion $o_i$ is assigned a reward $r_i$ via $R_\phi(o_i)$, normalized by the groups mean and standard deviation:

$$\widetilde{r}_i = \frac{r_i - \text{mean}\big(\{r_1, \ldots, r_G\}\big)}{\text{std}\big(\{r_1, \ldots, r_G\}\big)}.$$

All tokens in $o_i$ share the same advantage, $\hat{A}_{i,t} = \widetilde{r}_i$, for $t = 1, \ldots, |o_i|$. The second stage thus reduces to a standard GRPO update applied to the augmented prompt $q'$:

$$\mathcal{J}_{\text{iGRPO}}(\theta) = \mathbb{E}\Big[q \sim P(Q)\Big] \, \mathbb{E}\Big[\underbrace{\{r_i\}_{i=1}^N \sim \pi_{\theta_{\text{old}}}(\cdot \mid q)}_{\text{Stage 1}}, \; \hat{r} = \arg\max\{R_\phi(r_i)\}, \; \underbrace{\{o_j\}_{j=1}^G \sim \pi_{\theta_{\text{old}}}(\cdot \mid q')}_{\text{Stage 2}}\Big]$$

$$\times \frac{1}{G} \sum_{j=1}^G \frac{1}{|o_j|} \sum_{t=1}^{|o_j|} \Big[\min\Big(r_{j,t}(\theta) \, \hat{A}_{j,t}, \; \text{clip}\big(r_{j,t}(\theta), 1-\epsilon, 1+\epsilon\big) \, \hat{A}_{j,t}\Big) - \beta \, D_{\text{KL}}\big(\pi_\theta \| \pi_{\text{ref}}\big)\Big],$$

$$\tag{3}$$

where

$$r_{j,t}(\theta) = \frac{\pi_\theta(o_{j,t} \mid q', o_{j,<t})}{\pi_{\theta_{\text{old}}}(o_{j,t} \mid q', o_{j,<t})}.$$

and the other terms follow Section 3.1. Note that the only algorithmic difference from vanilla GRPO is that each group of completions is conditioned on an augmented prompt $q'$, which itself is derived by selecting the top-scoring Stage 1 completion. This provides a powerful *selffeedback* signal that helps the policy refine its solutions beyond the best candidate it generated in the initial pass.

By giving the policy access to its own high-quality draft as an in-context exemplar, iGRPO achieves systematic improvements in performance while incurring only minimal extra sampling overhead compared to standard GRPO. Empirically, we show that this feedback-driven refinement consistently boosts solution quality across a range of mathematical reasoning tasks and model architectures (see Section 4).

## 4 EXPERIMENTS

Our training data includes two datasets: MATH (Hendrycks et al., 2021) (7,500 step-by-step competition problems) and AceReason-Math (Chen et al., 2025b) (9,400 problems). All models are trained for one epoch, with a KL divergence loss coefficient of 0 and no entropy regularization. We use a learning rate of $1 \times 10^{-6}$ with a cosine schedule, generating eight completions per prompt (halved in each stage for iGRPO). The maximum prompt length is 1,024 tokens across all datasets. For all experiments, the completion length is capped at 4,096 tokens and the batch size at 1024. Although iGRPO runs about 17% slower than GRPO due to its second-stage refinement, its overall computational overhead remains modest and does not increase memory demand. This relative slowdown is justified by the consistent accuracy gains delivered across benchmarks, making the extra computation a practical tradeoff (see supplementary materials for details.)

We evaluate model performance on well-known mathematical benchmarks, including AIME24/AIME25, MATH500, AMC23, GSM8K, and Minerva Math. For all benchmarks we report Pass@1 accuracy. However, for AIME24/AIME25 the reported value is averaged over 64 runs to ensure robustness. For other benchmarks, an average of 8 runs are reported. For all evaluations, we use NVIDIA's NeMo-Skills framework[1] with decoding parameters such as a temperature of 0.6, top-p of 0.95, and generation length of 65,000.

## 5 EXPERIMENTS

### 5.1 CONTROLLED STUDIES

Table 1 reports a controlled comparison between *iGRPO*, vanilla GRPO, and two recent self-improvement baselines,namely Self-Verification (Zhang et al., 2025a) and Critique-GRPO (Zhang et al., 2025b), across several model families and parameter scales (7B, 8B, 14B). We interpret the results and discuss (i) how self-feedback interacts with model capacity and pretraining strength, and (ii) why the proposed two-stage refinement remains competitive with, and often superior to, more elaborate critique-based objectives.

**Generalist 8B model: large gains from self-feedback.** For the **Nemotron-H-8B-Base-8K** model, GRPO boosts average accuracy from 29.65% to 41.08%, and Self-Verification and Critique-GRPO reach 42.86% and 43.39%. iGRPO performs best at 45.04%, with clear gains on AIME25 (9.17%) and GSM8K (91.26%). The main advantage stems from avoiding the multi-task burden of self-judgment. Instead of requiring the model to verify or critique its own outputs, iGRPO simply conditions Stage 2 on the strongest Stage 1 draft, providing a clean exemplar and an easier refinement target for a capacity-limited generalist model.

**Stronger 7B reasoner: consistent but smaller margins.** For **DeepSeek-R1-Distill-Qwen-7B**, already strong at 61.93%, GRPO raises accuracy to 68.29%, and Self-Verification and Critique-GRPO reach 69.08% and 69.14%. iGRPO still performs best at 69.87%, with gains focused on multi-step tasks prone to near-miss errors (e.g., AIME24: 56.30%, AMC: 95.00%). Conditioning Stage 2 on the strongest Stage 1 draft steers the model toward a solid reasoning trajectory, requiring only minor fixes rather than learning a full self-critique, which explains the consistent advantage even for a strong 7B model.

**Math-specialized 7B models.** For **OpenMath-Nemotron-7B**, which is already strong at 74.83%, GRPO provides only a small improvement to 75.02%, while iGRPO increases performance to 76.07%. The largest gains appear on the harder benchmarks, such as AIME24, which rises from 73.28% to 74.79%, and AMC, which reaches 97.50%. Because the base model already produces high-quality solutions, Stage 1 often generates strong drafts, and Stage 2 refines them by reinforcing the most reliable reasoning patterns rather than introducing broad corrective signals.

---

[1] https://github.com/NVIDIA/NeMo-Skills

Table 1: Performance comparison of 7B, 8B, and 14B models across multiple mathematical reasoning benchmarks. Bold indicates the best and underlined the second best per column within each parameter bucket (7B, 8B, or 14B). Our method *iGRPO* rows are lightly shaded.

| Model | AIME25 | AIME24 | MATH500 | AMC | GSM8K | Minerva | Avg |
|---|---|---|---|---|---|---|---|
| Nemotron-H-8B-Base-8K | 6.20 | 8.65 | 61.23 | 43.21 | 41.02 | 17.60 | 29.65 |
| Nemotron-H-8B-Base-8K + GRPO | 7.78 | 9.01 | 73.13 | 45.10 | 81.93 | 29.56 | 41.08 |
| Nemotron-H-8B-Base-8K + Self-Verification (Zhang et al., 2025a) | 8.50 | 9.25 | 75.60 | 46.50 | 86.20 | 31.10 | 42.86 |
| Nemotron-H-8B-Base-8K + Critique-GRPO (Zhang et al., 2025b) | 8.42 | 9.15 | 76.05 | 46.80 | 88.40 | 31.50 | 43.39 |
| **Nemotron-H-8B-Base-8K + iGRPO** | **9.17** | **9.56** | **78.80** | **48.75** | **91.26** | **32.72** | **45.04** |
| DeepSeek-R1-Distill-Qwen-7B (Guo et al., 2025) | 38.60 | 54.40 | 92.80 | 90.00 | 92.00 | 39.10 | 61.93 |
| DeepSeek-R1-Distill-Qwen-7B + GRPO | 38.90 | 55.00 | 93.25 | 90.00 | 92.12 | 40.44 | 68.29 |
| DeepSeek-R1-Distill-Qwen-7B + Self-Verification (Zhang et al., 2025a) | 39.45 | 55.80 | 93.50 | 92.50 | 92.20 | 41.00 | 69.08 |
| DeepSeek-R1-Distill-Qwen-7B + Critique-GRPO (Zhang et al., 2025b) | 39.60 | 55.65 | 93.45 | 92.80 | 92.25 | 41.10 | 69.14 |
| **DeepSeek-R1-Distill-Qwen-7B + iGRPO (ours)** | **40.16** | **56.30** | **93.80** | **95.00** | **92.42** | **41.54** | **69.87** |
| OpenMath-Nemotron-7B (Moshkov et al., 2025) | 61.18 | 73.28 | 95.55 | 95.00 | 90.52 | 33.46 | 74.83 |
| OpenMath-Nemotron-7B + GRPO | 61.32 | 73.62 | 95.60 | 95.00 | 90.60 | 34.00 | 75.02 |
| **OpenMath-Nemotron-7B + iGRPO (ours)** | **62.45** | **74.79** | **96.00** | **97.50** | **90.75** | **34.94** | **76.07** |
| DeepSeek-R1-Distill-Qwen-14B (Guo et al., 2025) | 42.10 | 58.93 | 93.10 | 90.00 | 93.10 | 45.59 | 70.47 |
| DeepSeek-R1-Distill-Qwen-14B + GRPO | 43.70 | 60.26 | 93.10 | 91.20 | 93.50 | 46.00 | 71.29 |
| **DeepSeek-R1-Distill-Qwen-14B + iGRPO (ours)** | **45.52** | **64.06** | **94.00** | **93.45** | **94.00** | **47.06** | **73.02** |
| OpenMath-Nemotron-14B (Moshkov et al., 2025) | 61.18 | 73.28 | 95.55 | 95.00 | 94.01 | 33.46 | 75.41 |
| OpenMath-Nemotron-14B + GRPO | 64.53 | 74.79 | 96.00 | 95.00 | 94.40 | 35.70 | 76.73 |
| **OpenMath-Nemotron-14B + iGRPO (ours)** | **65.57** | **76.72** | **96.70** | **97.50** | **94.69** | **36.76** | **78.00** |

**Scaling to 14B parameters: persistent benefits on complex tasks.** At the 14B scale, the same trend holds. For **DeepSeek-R1-Distill-Qwen-14B**, average accuracy increases from 70.47% (base) to 71.29% with GRPO, and further to 73.02% with iGRPO, with the largest gains on difficult tasks (AIME24 improves from 58.93% to 64.06%). Similarly, **OpenMath-Nemotron-14B** rises from 75.41% (base) to 76.73% under GRPO and to 78.00% with iGRPO, including notable improvements on AIME25 (from 61.18% to 65.57%) and AIME24 (from 73.28% to 76.72%). These results show that even heavily pretrained math-oriented models continue to benefit from self-feedback.

**Scaling behavior and efficacy on complex tasks.** Across models, iGRPO improves GRPO by 1–4 points, with the largest gains on difficult reasoning tasks (AIME24/25, Minerva). As scale increases, gains shrink but remain meaningful: Stage 1 provides a strong draft, and Stage 2 refines ithelping smaller models find good modes and larger ones fix residual errors. Unlike Self-Verification or Critique-GRPO, which force the model to judge its own outputs and introduce noisy multi-task behavior, iGRPO uses an external reward and a validated best draft, giving a cleaner learning signal. This simple mechanism consistently strengthens GRPO on complex multi-step reasoning tasks.

## 5.2 GENERALIZATION TO A STRONGER BASE AND HARDER DATASET

Beyond the controlled comparisons in Table 1 (trained on the 7500problem MATH set), we test whether *iGRPO* also helps when (i) token horizons are longer, (ii) problems are more advanced, and (iii) the base model is already strong. We start from **OpenReasoningNemotron7B** and train on the **AceReason-Math** (Chen et al., 2025b) dataset, which contains harder competitionstyle items than MATH. We keep the *iGRPO* recipe identical to Section 4.

As shown in Fig. 2, *iGRPO* yields consistent gains over a strong baseline: **+1.52** on **AIME24** (85.62 vs. 84.10), **+1.78** on **AIME25** (79.64 vs. 77.86),

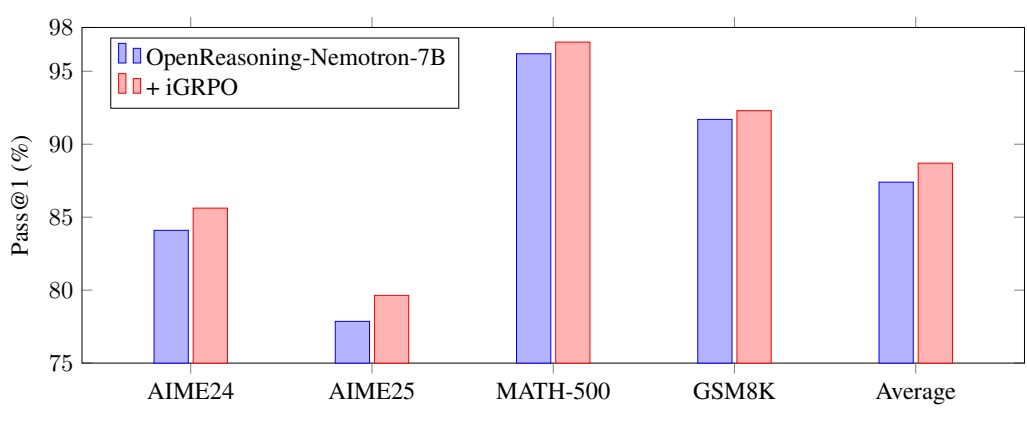

Figure 2: External validation on **OpenReasoning-Nemotron-7B** trained on AceReason-Math (Chen et al., 2025b) dataset. *iGRPO* consistently improves Pass@1 and concentrates gains on the harder competition benchmarks (AIME24/25).

**+0.80** on **MATH500** (97.00 vs. 96.20), and **+0.60** on **GSM8K** (92.30 vs. 91.70). The overall average improves by **+1.30** points (88.70 vs. 87.40). The largest gains appear on AIME24/25, aligning with the intuition that conditioning Stage 2 on the top Stage 1 draft is most helpful when deeper multistep reasoning is required. Within our 7B setting on AceReasonMath (Chen et al., 2025b) and evaluation protocol, these are the strongest AIME results we observe. Based on these results, *iGRPO* remains effective when training on a corpus with longer, more elaborate solutions. The improvements concentrate on AIME24/25, indicating that selffeedback particularly benefits difficult competition questions. In addition, even when starting from a capable 7B reasoner, *iGRPO* delivers consistent gains.

| Model | Avg |
|---|---|
| DeepSeek–R1–Distill–Qwen–7B + DAPO | 69.30 |
| **DeepSeek–R1–Distill–Qwen–7B + iDAPO (ours)** | **70.70** |
| DeepSeek–R1–Distill–Qwen–7B + GSPO | 69.20 |
| **DeepSeek–R1–Distill–Qwen–7B + iGSPO (ours)** | **70.60** |

Table 2: **Beyond GRPO:** layering our selffeedback refinement on DAPO and GSPO yields consistent gains (+1.4 points absolute in both cases) under matched training/eval settings. "Avg" is the macroaverage Pass@1 over the six benchmarks in Table 1.

# 6 ABLATION

**Beyond GRPO: Composing Self-Feedback with GRPO Variants.** We test whether our self-feedback refinement is *orthogonal* to recent GRPO variants DAPO (Yu et al., 2025) and GSPO (Zheng et al., 2025). Using the same setup as Table 1, we replace the Stage-2 GRPO update with DAPO or GSPO and apply our two-stage mechanism, producing *iDAPO* and *iGSPO*, with matched data, compute, and sampling budgets. Evaluation mirrors Table 1 ("Avg" = macro-average Pass@1 across AIME24/25, MATH500, AMC, GSM8K, Minerva). Self-feedback yields consistent gains: iDAPO improves over DAPO by **+1.4** (70.70 vs. 69.30), and iGSPO over GSPO by **+1.4** (70.60 vs. 69.20), paralleling iGRPO results. These results suggest iterative self-feedback complements optimization-centric methods: while dynamic/decoupled or sequence-level clipping address optimizer bias/variance, our Stage-1/Stage-2 process reshapes the *conditioning context* to emphasize high-quality solution modes. Because total sampling is fixed (e.g., $4+4$ vs. $8$), improvements arise not from extra decoding but from re-using the best draft as an in-context guide.

**Training Dynamics and Response Length.** As shown in Fig. 3 (a), we compare average rewards for iGRPO and GRPO at multiple checkpoints, observing that iGRPO consistently maintains a higher

reward throughout training. The iterative refinement in iGRPO ultimately yields a superior reward trajectory. In addition, as shown in Fig. 3 (b), we measure the response length over training steps and find that both methods exhibit nearly identical lengths, with GRPO producing slightly longer outputs on average. Notably, iGRPOs two-stage process does not manifest in lengthy completions but instead appears to refine solutions within a similar token budget. This indicates that the gains from iterative refinement arise more from improved response quality than from verbosity.

**Effect of KL Divergence Term.** We vary the coefficient $\beta \in \{0, 0.0001, 0.001, 0.01\}$ to examine how tightly the policy is regularized against the reference model. As shown in Fig. 4 (a), while $\beta = 0.0001$ achieves the highest overall score (70.23%), the difference among all settings is relatively small. The KL term, in principle, balances exploration with adherence to the current policy. However, given the marginal gains observed, setting $\beta = 0$ offers a simpler training pipeline without sacrificing significant performance. Hence, we use $\beta = 0$ to reduce overhead and maintain efficiency.

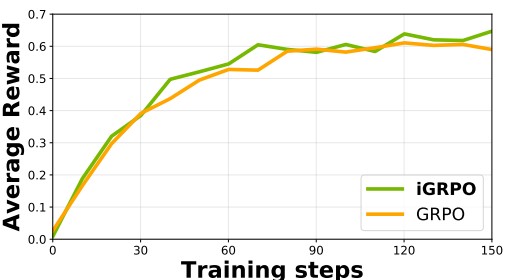

(a) Average training reward.

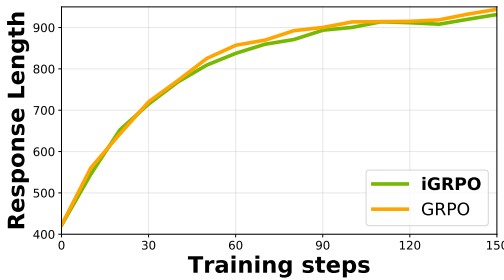

(b) Response length.

Figure 3: (a) Comparison of average rewards (b) response lengths (b) for GRPO vs. iGRPO.

**Effect of Number of Completions.** We study how the total number of completions in iGRPO affects performance, allocating $4, 8, 16,$ or $32$ completions evenly across the two stages. As shown in Fig. 4 (b) and the table, increasing from $4$ to $8$ completions gives a clear improvement, while gains beyond $8$ are modest. Larger budgets also increase training time and inference latency for minimal returns. Thus, we use $8$ total completions ($4$ per stage) as a practical balance.

# 7 CONCLUSION

We introduced Iterative GRPO (iGRPO), a simple two-stage self-feedback extension of GRPO that reliably refines completions and strengthens mathematical reasoning. Across controlled studies and a stronger base on AceReason-Math, iGRPO consistently surpasses GRPO, composes with DAPO and GSPO, and sets state-of-the-art results of 85.62% on AIME24 and 79.64% on AIME25 with modest additional cost. Future work will explore richer feedback selection, verifier-augmented rewards, and applications beyond mathematics.

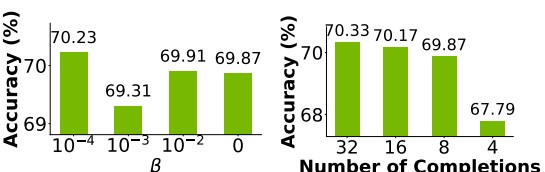

Figure 4: (a) $\beta$ and (b) number of generations.

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

APPENDIX

## A  POLICY GRADIENT DERIVATION

Below, we present a step-by-step derivation of the Iterative GRPO (iGRPO) policy gradient, showing how the two-stage sampling mechanism translates into a principled, value-function-free reinforcement learning update. We let $\pi_\theta$ be the current trainable policy, $\pi_{\theta_{\text{old}}}$ be the frozen policy used for sampling, and $R_\phi$ be an external reward function (or verifier). In this derivation, we adopt a *group-based advantage* without dividing by the standard deviation of within-group rewards or the completion length, thus circumventing sources of bias that can arise in other group-based RL objectives.

**Stage 1 (Self-Feedback).** Given a prompt $q$, we first sample a set of $N$ candidate completions $\{r_i\}_{i=1}^N$ from $\pi_{\theta_{\text{old}}}$:

$$r_i \sim \pi_{\theta_{\text{old}}}(\cdot \mid q), \quad i = 1, \dots, N. \tag{1}$$

We evaluate each candidate using the reward function: $\{R_\phi(r_i)\}_{i=1}^N$, and select the *top-scoring* draft:

$$\hat{r} = \arg\max_{1 \le i \le N} R_\phi(r_i). \tag{2}$$

We then form an augmented prompt $q'$ by concatenating $q$ and $\hat{r}$:

$$q' = [\, q; \, \langle\text{SelfFeedback}\rangle; \, \hat{r} \,].$$

This augmented prompt $q'$ captures an in-context example of the models own best prior response, facilitating iterative refinement.

**Stage 2 (Self-Reflection).** From $\pi_{\theta_{\text{old}}}$, we sample a *group* of $G$ completions conditioned on $q'$:

$$\{\, o_j \,\}_{j=1}^G \sim \pi_{\theta_{\text{old}}}(\cdot \mid q').$$

Let $\{R_\phi(o_1), \dots, R_\phi(o_G)\}$ be the scalar rewards for these completions. Define the group-based advantage for the $j$-th completion as:

$$\widetilde{A}_j = R_\phi(o_j) - \frac{1}{G} \sum_{k=1}^G R_\phi(o_k). \tag{3}$$

Notably, we do *not* divide by the standard deviation of the group rewards and do *not* normalize by $|o_j|$. This choice avoids injecting additional factors that can distort the magnitude of policy gradients, thereby preserving the directness of the reward-to-advantage mapping.

**Policy Gradient for iGRPO.** We now derive the policy gradient objective that merges the self-feedback mechanism (Stage 1) with standard group-based reinforcement learning (Stage 2). Because the prompt $q'$ itself depends on $\hat{r}$, which is sampled from $\pi_{\theta_{\text{old}}}$, the second stage can be viewed as an RL update *conditioned* on the random augmented prompt. Formally, let $\{o_j\}_{j=1}^G \sim \pi_{\theta_{\text{old}}}(\cdot \mid q')$ and denote $|o_j|$ as the token length of the $j$-th completion. Each token $o_{j,t}$ (for $t = 1, \dots, |o_j|$) is drawn from $\pi_{\theta_{\text{old}}}(\cdot \mid q', o_{j,<t})$. By the principle of REINFORCE and advantage weighting, the gradient of the objective w.r.t. the parameters $\theta$ of the current policy $\pi_\theta$ can be expressed as:

$$\nabla_\theta \, \mathcal{J}_{\text{iGRPO}}(\theta) = \mathbb{E}_{q \sim P(\mathcal{Q})} \, \mathbb{E}_{\{r_i\}_{i=1}^N \sim \pi_{\theta_{\text{old}}}(\cdot \mid q)} \, \mathbb{E}_{\hat{r}=\text{top\_scoring}} \, \mathbb{E}_{\{o_j\}_{j=1}^G \sim \pi_{\theta_{\text{old}}}(\cdot \mid q')}$$

$$\times \frac{1}{G} \sum_{j=1}^G \sum_{t=1}^{|o_j|} \nabla_\theta \, \log \pi_\theta\big( o_{j,t} \mid q', o_{j,<t} \big) \, \widetilde{A}_j. \tag{4}$$

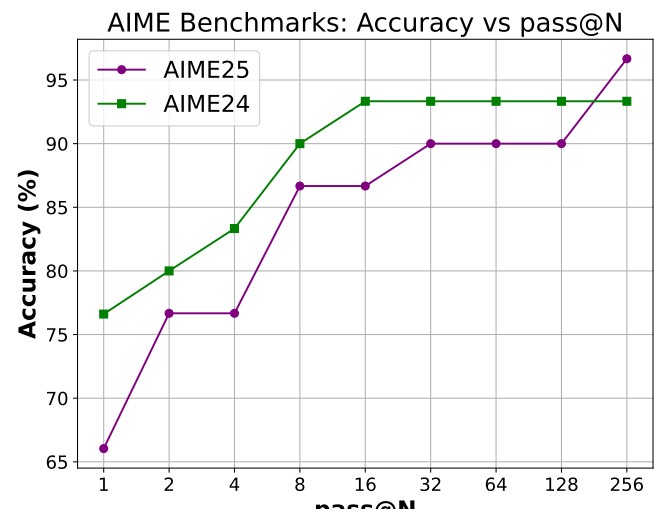

Figure S.1: Performance of iOpenMath-Nemotron-14B across various pass@N settings for AIME24 and AIME25. Both benchmarks exhibit increasing accuracy with higher $N$, though AIME24 quickly stabilizes at 93.33% by $N = 16$, whereas AIME25 continues to rise until reaching 96.67% at $N = 256$.

Table S.1: Performance comparison of OpenMath-Nemotron-14B and iGRPO-enhanced variant.

| Model | AIME25 | AIME24 | MATH500 | AMC | GSM8K | Minerva | Avg |
|---|---|---|---|---|---|---|---|
| OpenMath-Nemotron-14B (Moshkov et al., 2025) | 61.18 | 73.28 | 95.55 | 95.00 | 94.01 | 33.46 | 75.41 |
| **OpenMath-Nemotron-14B + iGRPO** | **66.04** | **76.61** | **96.90** | **97.50** | **94.16** | **38.24** | **78.24** |

Here, $\widetilde{A}_j$ does not depend on individual token steps, so it factors out of the summation over $t$. Introducing the usual PPO-style clipping mechanism for better stability, we define

$$r_{j,t}(\theta) \;=\; \frac{\pi_\theta\big(o_{j,t} \,|\, q', \, o_{j,<t}\big)}{\pi_{\theta_{\mathrm{old}}}\big(o_{j,t} \,|\, q', \, o_{j,<t}\big)},$$

and write the resulting *unclipped* versus *clipped* objective terms. Hence, the final iGRPO update objective becomes:

$$\mathcal{J}_{\mathrm{iGRPO}}(\theta) \;=\; \mathbb{E}\Big[q \sim P(\mathcal{Q})\Big] \, \mathbb{E}\Big[\{r_i\}_{i=1}^N \sim \pi_{\theta_{\mathrm{old}}}(\cdot\,|\,q)\Big] \, \mathbb{E}\Big[\hat{r} = \arg\max_i R_\phi(r_i)\Big] \, \mathbb{E}\Big[\{o_j\}_{j=1}^G \sim \pi_{\theta_{\mathrm{old}}}(\cdot\,|\,q')\Big]$$

$$\times \; \frac{1}{G} \sum_{j=1}^{G} \sum_{t=1}^{|o_j|} \min\Big(r_{j,t}(\theta)\,\widetilde{A}_j, \; \mathrm{clip}\big(r_{j,t}(\theta), 1-\epsilon, 1+\epsilon\big)\,\widetilde{A}_j\Big). \qquad (4)$$

# B    SCALING OPENMATH-NEMOTRON-14B WITH iGRPO

To further validate the effectiveness of iGRPO, we trained the OpenMath-Nemotron-14B with iGRPO on large scale dataset of OpenR1-Math-220k (R1, 2024) which consists of 220,000 math problems with reasoning traces from DeepSeek-R1. For this study, we use 94,000 examples. As shown in Table S.1, the reasoning

Table S.2: Hyperparameters for training 7B models with iGRPO using 2 nodes each using 8 Œ NVIDIA A100 GPUs. One node is entirely reserved for vllm.

| Parameter | Value |
|---|---|
| bf16 | true |
| attn_implementation | flash_attention_2 |
| use_vllm | true |
| vllm_gpu_memory_utilization | 0.85 |
| gradient_accumulation_steps | 8 |
| gradient_checkpointing | true |
| learning_rate | 1e-06 |
| lr_scheduler_type | cosine_with_min_lr |
| lr_scheduler_kwargs | min_lr_rate: 0.1 |
| warmup_ratio | 0.1 |
| num_train_epochs | 1 |
| per_device_train_batch_size | 16 |
| max_completion_length | 4096 |
| num_generations | 4 |
| temperature | 0.7 |
| reward_funcs | accuracy, format |
| reward_weights | 1.0, 1.0 |

performance of the model trained with iGRPO is significantly improved, achieving an impressive AIME25 score of 66.04%.

### B.1 ANALYSIS OF PASS@N ON AIME BENCHMARKS

Figure S.1 shows how the accuracy of iOpenMath-Nemotron-14B evolves when increasing the number of attempts $N$ on the AIME24 and AIME25 benchmarks. As expected, we see consistent gains in performance as $N$ grows, indicating that the model can generate correct solutions among multiple sampled responses even if the top-1 guess is sometimes incorrect. Although our base SFT model already shows strong results at pass@1, the additional RL fine-tuning appears to capitalize on multi-sample scenarios. For example, on AIME25, the model improves from 66.04% (pass@1) to 86.67% at pass@8, demonstrating the effectiveness of producing multiple solutions for challenging competition problems.

Despite these gains, there are clear saturation points. AIME24 converges to its best score of 93.33% by $N = 16$, with no further improvement at higher $N$. On the contrary, the performance of AIME25 continues to improve even at high values of $N$. While the performance seems to plateau briefly at 90.00% between $N = 32$ and $N = 128$, it eventually increases to 96.67% at $N = 256$. This contrast suggests that certain problem distributions, especially in AIME25, may benefit from a larger number of sampled attempts, whereas others, such as those in AIME24, can be adequately solved with fewer solution attempts.

## C HYPERPARAMETER SETUP

We conduct ablation studies on both 7B and 14B model variants, spanning core architectures such as DeepSeek-R1-Distill-Qwen and OpenMath-Nemotron. All models are trained for one epoch across two

datasets: (1) the MATH dataset (Hendrycks et al., 2021) of 7,500 step-by-step problems, and (2) AceReasonMath (Chen et al., 2025b) dataset (9400 problems). Our training uses a KL divergence loss coefficient of 0. We set a cosine learning rate schedule (minimum rate of 0.1) with a base learning rate of $1 \times 10^{-6}$. We use 8 rollouts for all experiments.

Table S.2 lists the concrete hyperparameters for training 7B iGRPO models. Notably, we run on 2 nodes with $8 \times$ NVIDIA A100 GPUs each, and one of these nodes is fully allocated to vLLM for generation. We keep a global batch size of 128, with a per-device batch size of 16 and a gradient accumulation step size of 8. For the 14B models, we scale out to 5 nodes of $8 \times$ NVIDIA A100 GPUs each (again, one node reserved for vLLM) and reduce the per-device batch size to 4 (maintaining the same global batch size of 128). In both 7B and 14B setups, we continue to use bfloat16 precision and the FlashAttention-2 kernel. The temperature is set to 0.7 for generation, and we apply two reward functions (accuracy and format) each with weight 1.0. This configuration provides a balanced trade-off between training stability, throughput, and alignment with complex mathematical reasoning tasks.

**Prompt:** We use the following prompt for training model with iGRPO.

> **Prompt**
>
> You are a helpful AI Assistant that provides well-reasoned and detailed responses. First think through your reasoning as an internal monologue, then give the answer: <think></think><answer></answer>. If the prompt has a <SelfFeedback> tag, then treat all text that appears after as useful feedback on your previous response. Incorporate that feedback to refine, improve, and extend your answer.

# D MEMORY AND THROUGHPUT COMPARISONS

## D.1 SETUP

To evaluate the resource utilization of our training setup, we replicate the exact environment and conditions under which our 7B models are typically trained. Specifically, we use DeepSeek-R1-Distill-Qwen-7B as our base model, which serves as a representative checkpoint for measuring throughput and memory consumption when training on the MATH dataset. Our training configuration employs a per-device batch size of 16, along with a global gradient accumulation step of 8, allowing us to effectively simulate heavier loads without exceeding GPU memory constraints. Additionally, we use a maximum completion length of 2048 tokens to benchmark model performance. We run experiments on two nodes, each equipped with 8 Œ NVIDIA A100 GPUs. One node is dedicated to vllm generation, ensuring that inference or generation processes do not interfere with the primary training workload, while the other node is reserved exclusively for model training.

We measure peak memory usage by periodically querying the GPU memory allocator for the maximum memory allocation that has occurred since the start of training. Specifically, at the beginning of training, we reset the peak memory statistics, and then after each iteration, we retrieve the current peak memory usage in bytes. We convert this value to gigabytes for readability and log it alongside other training metrics. To measure throughput, we track the total number of samples processed over time. We calculate this by multiplying the current global step by both the per-device batch size and the number of devices used in data parallelism. Dividing this product by the elapsed training time in seconds yields the throughput, expressed as samples processed per second. This real-time monitoring of memory and throughput allows us to evaluate hardware utilization efficiency, identify possible bottlenecks, and compare different training configurations in a consistent and quantifiable manner.

## D.2 Measurements

Table S.3 presents measured GPU usage and training throughput under iGRPO vs GRPO. Despite the two-stage nature of iGRPO, its peak memory usage of 54.9349,GB closely matches GRPOs 54.9286,GB,a difference of roughly 0.0063,GB, which is practically negligible. This matches our theoretical expectation that the self-feedback mechanism adds minimal overhead, validating the feasibility of integrating iterative refinements even under constrained resource budgets.

Regarding throughput, iGRPO processes 0.34,samples/s compared to GRPOs 0.41,samples/s, reflecting a mild slowdown tied to the additional round of generation. Crucially, this is neither an order-of-magnitude nor a large factor reduction. Instead, it shows that iGRPOs second-stage refinement imposes only a modest computational cost. In summary, these measurements confirm our claims that iGRPO can be implemented with little additional overhead, supporting it as a practical strategy for enhancing mathematical reasoning performance without compromising resource efficiency.

Beyond instantaneous throughput, we also report full training cost measured in total GPU hours. Under the same compute budget and eight generations per prompt, GRPO requires 83.3 GPU hours while iGRPO uses 94.1 GPU hours, which corresponds to roughly a 13% increase in wall-clock training time. This overhead arises from the sequential Stage 1 plus Stage 2 decoding but does not demand more GPUs or additional memory capacity, since peak usage remains essentially unchanged. Given that this modest time increase delivers several-point gains on AIME24 and AIME25 and enables our 7B models to reach state-of-the-art performance, we view the tradeoff between 13% extra training time and substantially higher reasoning accuracy as a favorable and practical value proposition in real deployments.

| Method | Peak Memory (GB) | Throughput (Samples/sec) | Total GPU Hours |
|--------|------------------|--------------------------|-----------------|
| GRPO   | 54.9286          | 0.41                     | 83.3            |
| iGRPO  | 54.9349          | 0.34                     | 94.1            |

Table S.3: Empirical memory usage and throughput on an 80 GB A100 setup. iGRPOs two-stage approach yields near-identical peak memory usage and only a minor decrease in throughput compared to GRPO. The last column reports total GPU hours for a full training run, showing that iGRPO adds only a modest 13% time overhead relative to GRPO.

# E   Additional Ablation Studies

## E.1 Reward model design and limitations

In all main experiments we use a simple outcome reward that is computed by a rule based verifier, rather than by a learned reward model. Given a problem instance (q) with reference answer (a) and a model completion (o), the verifier extracts the final answer from (o) and compares it with (a) under the official competition scoring scheme. If the solution is correct the reward is ($r(q,o) = 1$), otherwise ($r(q,o) = 0$). This scalar is treated as the reward for the entire reasoning trace and is used both to compute group normalized advantages in GRPO and iGRPO and to select the best Stage 1 draft in iGRPO. Since the reward is rule based and deterministic there is no separate training phase for the reward model and there is no risk of reward drift across iterations.

This choice has two advantages. First, the mapping from solution quality to reward is transparent and easy to inspect, which simplifies debugging and ablation studies. Second, the outcome reward is perfectly aligned with the benchmark metrics that we ultimately care about because it implements the same grading rules that define Pass@1. However, the design also introduces clear limitations. The reward is sparse and assigns no credit to partially correct derivations that contain the correct strategy but fail on a late algebraic or numeric

Table S.4: Effect of replacing the rule-based outcome reward with a GPT-5 generative judge inside iGRPO. The base model is DeepSeek R1 Distill Qwen 7B trained on MATH and we report Pass@1 in percent.

| Benchmark | iGRPO (rule based) | iGRPO (GPT-5 judge) | Δ |
|---|---|---|---|
| AIME25 | 40.16 | **41.12** | +0.96 |
| AIME24 | 56.30 | **57.45** | +1.15 |
| MATH500 | 93.80 | **94.20** | +0.40 |
| AMC | 95.00 | **96.25** | +1.25 |
| GSM8K | 92.42 | **92.95** | +0.53 |
| Minerva | 41.54 | **42.88** | +1.34 |
| Average | **69.87** | **70.81** | +0.94 |

step. In such cases all completions in a group can receive zero reward and the normalized advantage in GRPO or iGRPO collapses to zero, which reduces the effective learning signal to the KL regularizer.

The two stage structure of iGRPO partially mitigates this sparsity. Even under a binary reward the group relative normalization highlights the highest scoring samples inside each Stage 2 group. When some completions are correct and others are not, the correct ones receive positive advantage and are reinforced, while the incorrect ones are suppressed. Nevertheless, if the reward mechanism cannot distinguish between nearly correct and completely incorrect solutions then iGRPO cannot exploit the quality differences between them. This motivates exploring richer reward models that can provide dense feedback on the entire reasoning trace and not only on the final boxed answer.

## E.2  GENERATIVE JUDGE STUDY

Generative reward models such as GPT 5 integrate naturally into iGRPO. Conceptually, they are used as judges that read the full problem, the models chain of thought, and the final answer, and then return a scalar score ($r \in [0, 1]$) that summarizes overall solution quality. Within iGRPO this scalar plays exactly the same role as the rule based outcome reward. It is used to rank the Stage 1 drafts and select the best self feedback completion, and it is used again to compute group normalized advantages for the Stage 2 GRPO update. In future work we also plan to use short textual critiques from such judges as additional self feedback in the Stage 2 prompt, but here we only exploit the scalar signal.

To study the impact of reward model quality we run an additional experiment with DeepSeek R1 Distill Qwen 7B trained on MATH. We keep the iGRPO training recipe fixed and compare two reward variants. The first variant uses the rule based outcome reward ($r \in [0, 1]$) described above. The second variant replaces the rule based checker with GPT 5 acting as a generative judge that assigns a dense score ($r \in [0, 1]$) to each solution. Table S.4 reports Pass@1 on the same six benchmarks as in Table 1.

We observe consistent improvements from using the generative judge, with the largest gains on AIME24, AIME25, and Minerva. These benchmarks contain many problems where the model identifies the right high level strategy but makes a minor arithmetic or transcription error near the end of the solution. The rule based checker treats such solutions as entirely incorrect and assigns reward zero, so they cannot serve as self feedback and they do not contribute positive advantage. In contrast, GPT 5 often assigns an intermediate score to these partially correct traces. Under iGRPO these partially correct solutions then survive the Stage 1 selection step and are used as feedback exemplars, and the Stage 2 update refines them into fully correct answers. This explains why the gains are largest on benchmarks that emphasize long multi step reasoning.

| Method | Avg Pass@1 | Improvement |
|---|---|---|
| RLOO | 69.15 | - |
| iRLOO (ours) | 70.62 | +1.47 |

Table S.5: Comparison between RLOO and iRLOO under the same 7B setting, dataset, and compute budget as in Table 1. The reported numbers are macro averaged Pass at 1 over AIME24, AIME25, MATH500, AMC, GSM8K, and Minerva.

At the same time, generative reward models come with their own limitations. Their scores can be biased, noisy, or inconsistent across prompts, and they may favor certain stylistic features of solutions that are not directly related to correctness. Because GRPO and iGRPO only depend on the relative ranking of completions inside each group, they are robust to moderate noise and benefit whenever the judges scores are positively correlated with true correctness. However, a very poor or systematically biased generative judge could still misguide training. This suggests that more advanced uses of textual critiques and cross checked ensembles of judges are promising directions. Overall, the GPT 5 study supports our claim that iGRPO is compatible with both rule based and generative rewards and that its performance improves as the reward signal becomes more informative.

### E.3 RLOO COMPARISON AND STRONGER BASELINES

RLOO (Kool et al., 2019) is a strong variant of REINFORCE that reduces variance by subtracting a leave one out baseline from each completion reward. We keep the 7B setting, training data, and sampling budget identical to our main GRPO experiments. The total number of completions per prompt is fixed, and in the iterative variant half of the samples are used for Stage 1 and half for Stage 2, exactly as in iGRPO. Table S.5 summarizes the macro average Pass at 1 across the same six benchmarks.

In our setup RLOO alone slightly outperforms GRPO, which is consistent with its lower variance and unbiased estimator, yet adding the self feedback stage still gives a further gain of 1.47 points. Together with the DAPO and GSPO ablations, these results support the view that the two stage refinement is complementary to the choice of advantage estimator. Methods such as DAPO, GSPO, and RLOO focus on how the gradient is constructed from a fixed batch of trajectories, while iGRPO, iDAPO, iGSPO, and iRLOO modify the conditioning context by feeding the best Stage 1 draft into Stage 2. This conditioning shift steers the model toward higher reward solution modes before gradient computation, and it does so without increasing the total number of generated trajectories.

The same mechanism also remains compatible with stronger reward sources. Generative reward models such as GPT 5 can replace the rule based checker and provide scalar scores for each completion. iGRPO only requires a scalar reward per sample and the identity of the best Stage 1 draft, so a generative judge can be dropped in without changing the optimization logic. In future work the textual critique produced by such a judge can be concatenated with the best draft inside the Stage 2 prompt, which would allow the self feedback signal to include both an improved trajectory and a short explanation of how to fix earlier mistakes.

Overall, the additional comparisons with DAPO, GSPO, and RLOO show that the modest looking gains over plain GRPO in Table 1 persist when iGRPO is applied on top of stronger RL baselines. The iterative self feedback design therefore provides a simple and general recipe that can be composed with a range of group based policy gradient algorithms under similar training conditions.

Table S.6: Per-token policy entropy in nats for DeepSeek-R1-Distill-Qwen-7B trained on MATH. Initial values are measured at the first training step. Mid-training values are measured at roughly thirty percent of total steps. Final values are measured at the end of training.

| Method | Initial | Mid training | Final |
|--------|---------|--------------|-------|
| GRPO   | 2.45    | 0.42         | 0.42  |
| iGRPO  | 2.45    | 0.48         | 0.44  |

### E.4 ENTROPY ANALYSIS

We further analyzed the training dynamics by tracking the per-token Shannon entropy of the policy distribution during reinforcement learning. Entropy provides a scalar summary of the model uncertainty and exploration behavior at each decoding step.

For a given decoding step $t$ with context $h_t$ and vocabulary $\mathcal{V}$, we define the entropy $\mathcal{H}$ of the policy $\pi_\theta$ using the natural logarithm so the information is measured in nats:

$$\mathcal{H}\big(\pi_\theta(\cdot \mid h_t)\big) = - \sum_{w \in \mathcal{V}} \pi_\theta(w \mid h_t) \ln \pi_\theta(w \mid h_t). \tag{5}$$

In practice, we compute this quantity from the log-softmax of the logits (base $e$) and average it over all valid completion tokens in the batch. This produces a single scalar per training step that we use as an indicator of exploration.

We apply this analysis to DeepSeek-R1-Distill-Qwen-7B trained on the MATH dataset. Both GRPO and iGRPO start from the same initialization with an average entropy of $2.45$ nats. Under GRPO, the entropy quickly collapses and reaches $0.42$ nats within the first thirty percent of training steps. Under iGRPO, we observe a more gradual and sustained decay. The entropy stays around $0.48$ nats in the mid-training regime, then converges to $0.44$ nats by the end of training.

These measurements suggest that the self-feedback mechanism in iGRPO acts as a mild regularizer on the policy distribution. It helps maintain slightly higher diversity in the token-level predictions during the bulk of training. This additional exploration is preserved without preventing convergence, since the entropies of GRPO and iGRPO are close at the end of training. The behavior aligns with our empirical findings on reasoning benchmarks where iGRPO shows improved performance on complex problems that benefit from richer intermediate exploration.

## F IN-CONTEXT LEARNING DISTINCTION

Although the iGRPO update rule ultimately uses the same GRPO surrogate loss as in Equations equation 1 and equation 3, the training dynamics it induces differ fundamentally from both static in-context learning and standard data augmentation. The key distinction lies not in modifying the gradient estimator but in modifying the *rollout distribution* and the *conditioning structure* of the policy during training.

Standard GRPO trains a policy to model the mapping

$$P_\theta(a \mid q)$$

based on an on policy distribution of prompts $q$ and single stage completions. In contrast, iGRPO explicitly trains a self correction mapping

$$P_\theta\big(a \mid q, \hat{r}_\pi\big),$$

where $\hat{r}_\pi$ is the highest reward completion sampled from the current policy for the same prompt. This feedback signal is not drawn from a fixed dataset nor from an external retrieval pool. It is generated online by the evolving policy itself and selected according to the reward model. As a result, the augmented prompt

$$q' = [q; \langle \text{SelfFeedback} \rangle; \hat{r}_\pi]$$

is a random variable whose distribution changes as the policy improves, directly coupling the conditioning context to the model's own evolving capabilities.

This dynamic feedback mechanism differentiates iGRPO from classical in-context learning. In standard ICL, conditioning examples are fixed once chosen and remain static throughout training. Under iGRPO, the feedback used in Stage 2 changes continually, forming a moving target that co evolves with the policy. As the model improves, the distribution over $\hat{r}_\pi$ shifts toward more accurate and more structured reasoning traces. The policy is repeatedly trained to refine and surpass its own strongest drafts, producing a sequence of self generated curricula in which refinement tasks naturally increase in difficulty. This type of online curriculum cannot be achieved with static ICL or offline augmentation.

iGRPO also differs from data augmentation. A data augmentation interpretation would treat $(q, \hat{r}_\pi)$ as a fixed additional datapoint that can be precomputed. In iGRPO, however, $\hat{r}_\pi$ is recomputed at every iteration under the same sampling budget used by GRPO. The algorithm restructures the rollout pipeline by splitting sampling into two stages, selecting a best Stage 1 candidate, and conditioning Stage 2 on that candidate. This directly alters the state visitation distribution. Stage 2 completions begin from states that already encode high quality drafts, and the update focuses on improving these drafts rather than generating solutions from scratch. The policy is therefore trained to use self feedback as a scaffold for reasoning.

Empirically, this modification produces substantial gains on challenging multi-step reasoning tasks. The effectiveness of iGRPO stems not from altering the GRPO loss itself but from altering the distribution of contexts on which that loss is applied, enabling reinforcement learning to exploit a dynamic self refinement mechanism that static ICL or data augmentation cannot provide.

## G  SELF-REFLECTION IN IGRPO

In iGRPO the term self reflection refers to an explicit two stage procedure in which the policy first explores a space of candidate solutions and then uses the best available outcome as an in context guide for refinement. This section formalizes that notion and clarifies how it differs from simply conditioning on an arbitrary previous output.

**Two stage self feedback mechanism**    For each prompt $(q)$ the procedure begins with an active exploration stage. The policy $(\pi_{\theta_{\text{old}}})$ samples a set of candidate completions

$$r_i \sim \pi_{\theta_{\text{old}}}(\cdot \mid q) \quad i = 1, \ldots, N$$

and an external reward model $(R_\phi)$ evaluates every candidate. The system then selects

$$\hat{r} = \arg \max_{i \in \{1, \ldots, N\}} R_\phi(r_i)$$

which we refer to as the self feedback draft. This selection step imposes a preference ordering over the models own outputs and ensures that only the strongest trajectory is used as feedback.

The second stage uses this selected draft as an explicit anchor for refinement. The original prompt is augmented to

$$q' = [q \text{ "<SelfFeedback>" } \hat{r}]$$

and the policy again samples a group of completions

$$o_j \sim \pi_{\theta_{\text{old}}}(\cdot \mid q') \quad j = 1, \ldots, G.$$

Group relative rewards on $(\{o_j\}_{j=1}^{G})$ then drive the GRPO style update as in Equation equation 3. In this way Stage two is explicitly conditioned on the best outcome from Stage one and is optimized to move probability mass toward completions that not only resemble $(\hat{r})$ in quality but can surpass it under the reward model.

**Interaction with the reward model**  A key property of this construction is that self reflection is always mediated by an external reward signal. The same policy does not both generate and evaluate its outputs. Instead, generation in both stages is performed by $\pi_{\theta_{\text{old}}}$, while evaluation and ranking are handled entirely by $R_\phi$. This division of roles reduces the risk of self-confirmation loops in which a model might otherwise reinforce its own unchecked preferences. The self-feedback draft is promoted to the conditioning context only when the reward model validates it as high quality. In this way, the two-stage mechanism implements an iterative refinement loop. The model first produces multiple drafts, then identifies the strongest one according to $R_\phi$, and finally updates its behavior using completions conditioned on this high-quality exemplar and optimized to achieve even higher reward.

