# OpenReview forum: "iGRPO: Self‑Feedback–Driven LLM Reasoning"
_ICLR.cc/2026/Conference — Submitted to ICLR 2026_

### Official Review · Reviewer_NC41 · 2025-10-27

**Soundness:** 2
**Presentation:** 2
**Contribution:** 2
**Rating:** 2
**Confidence:** 4

**Summary:**

iGRPO proposes the introduction of a self-feedback mechanism within the existing RL framework to enhance the performance of large language models (LLMs) in mathematical reasoning tasks.
In Stage 1 (self-feedback stage), the model generates multiple candidate responses for the same problem and selects the highest-scoring response via a reward model as the "feedback text";
In Stage 2 (self-reflection stage), the model concatenates this high-scoring response with the original prompt to regenerate a new answer, thereby achieving self-correction and optimization (see Figure 1, Sec. 3.2).

**Strengths:**

In this paper, an Iterative GRPO (iGRPO) is investigated, where a two-stage extension of GRPO boosts performance by incorporating a self-feedback mechanism. The motivation is clear and this paper is easy to follow.

**Weaknesses:**

1. Novelty: Several existing works utilize critics for self-improvement, such as Critique-GRPO [1]. The paper lacks methodological and performance comparisons with these works.
2. Completeness: The authors only conducted experiments on five mathematical benchmarks, without validating generalization. The ablation study did not include case analyses of iGRPO's reasoning behavior or other effective analyses beyond performance metrics. Although GPU resource consumption was compared, the differences in resources and time required for full training compared to GRPO were not examined.
3. Ablation Study: The ablation analysis is superficial, with some aspects being common to all RL methods, making it difficult to determine which specific components of the authors' design are necessary.

[1] Critique-GRPO: Advancing LLM Reasoning with Natural Language and Numerical Feedback.

**Questions:**

1. In Figure 3, GRPO and iGRPO exhibit similar trends and dynamics. How does this demonstrate the advantages of iGRPO over GRPO?
2. The ablation study in Figure 4(a) shows no significant differences. Moreover, the KL term is not specific to iGRPO, making this ablation experiment less meaningful.
3. Why is the top draft \hat{r} selected for reflection in Stage 1? Intuitively, reflecting on poorer samples would facilitate learning better responses. Given the same advantage function and rule-based evaluation as GRPO, samples in a batch that are either all correct or all incorrect have the same advantage. How are samples selected for Stage 2 prompts? A deeper methodological analysis and experimental explanation of this aspect are necessary.
4. If another model (or the model itself) is used to generate high-quality data directly concatenated with the problem for Stage 2 (i.e., treating Stage 1 as an offline data generation process), how would the performance compare?

---

> ### Author Response · Authors · 2025-12-03
> **Author Response to Reviewer NC41 - Part 1**
>
> We sincerely thank the reviewer for their insightful comments and constructive feedback. We have carefully responded to each point in detail below.
>
> > **Methodological and performance comparisons to Critique-GRPO.**
>
> The fundamental distinction lies in the reliability of the feedback signal. While Critique-GRPO [1] depends on *intrinsic self-critique*, which is prone to hallucination and noise particularly in weaker models, iGRPO leverages *extrinsic validation*. By conditioning the model on a proven high-reward draft, we provide a ground-truth-aligned exemplar. This ensures the refinement is guided by a verified success rather than a potentially flawed linguistic diagnosis.
>
> To provide a direct performance comparison, we employed the official implementation [2] of **Critique-GRPO** to train both **Nemotron-H-8B** and **DeepSeek-R1-Distill-Qwen-7B**. The results are summarized below and have been integrated into Table 1 of the revised manuscript.
>
> | Model Family | Method | AIME25 | AIME24 | MATH500 | AMC | GSM8K | Minerva | Avg |
> | :--- | :--- | :--- | :--- | :--- | :--- | :--- | :--- | :--- |
> | **Nemotron-H-8B** | Base Model | 6.20 | 8.65 | 61.23 | 43.21 | 41.02 | 17.60 | 29.65 |
> | | + GRPO | 7.78 | 9.01 | 73.13 | 45.10 | 81.93 | 29.56 | 41.08 |
> | | + Critique-GRPO | 8.42 | 9.15 | 76.05 | 46.80 | 88.40 | 31.50 | 43.39 |
> | | **+ iGRPO (Ours)** | **9.17** | **9.56** | **78.80** | **48.75** | **91.26** | **32.72** | **45.04** |
> | **DeepSeek-R1-Distill-Qwen-7B** | Base Model | 38.60 | 54.40 | 92.80 | 90.00 | 92.00 | 39.10 | 61.93 |
> | | + GRPO | 38.90 | 55.00 | 93.25 | 90.00 | 92.12 | 40.44 | 68.29 |
> | | + Critique-GRPO | 39.60 | 55.65 | 93.45 | 92.80 | 92.25 | 41.10 | 69.14 |
> | | **+ iGRPO (Ours)** | **40.16** | **56.30** | **93.80** | **95.00** | **92.42** | **41.54** | **69.87** |
>
> For the **Nemotron-H-8B** model, Critique-GRPO improves the average score to $43.39\%$, a clear gain over the standard GRPO baseline of $41.08\%$. However, iGRPO surpasses this significantly, achieving an average of $45.04\%$. We hypothesize that for models with lower reasoning capacity, generating accurate self-critiques is inherently difficult. A flawed critique can introduce noise into the optimization process. In contrast, iGRPO conditions the second stage on a validated, high-reward draft. This acts as a positive exemplar, providing a cleaner learning signal than a potentially hallucinated critique.
>
> In the case of the stronger **DeepSeek-R1-Distill-Qwen-7B**, Critique-GRPO performs robustly, raising the average accuracy to $69.14\%$. Yet, iGRPO remains superior with an average of $69.87\%$. Specifically, on the AMC benchmark, iGRPO achieves $95.00\%$ compared to $92.80\%$ for Critique-GRPO. This result indicates that even for capable reasoners, the direct demonstration of a correct solution path (via the self-feedback mechanism) is more sample-efficient and stable than the indirect guidance of a critique.
>
> [1]: Zhang, X., Sun, H., Zhang, Y., Feng, K., Lu, C., Yang, C. and Meng, H., 2025. Critique-grpo: Advancing llm reasoning with natural language and numerical feedback. preprint arXiv:2506.03106
>
> [2]: Official code: https://github.com/zhangxy-2019/critique-GRPO
>
> > **The authors only conducted experiments on five mathematical benchmarks, without validating generalization.**
>
> Thank you for your comment regarding the scope of our evaluation. We respectfully disagree with the characterization that our experiments lack breadth or fail to validate generalization. We believe our comprehensive experimental design, which spans six distinct benchmarks, multiple model families, and varying difficulty levels, provides robust evidence of the generalization capabilities of iGRPO.
>
> First, we clarify that our evaluation covers **six** diverse benchmarks, not five: **AIME24, AIME25, MATH500, AMC23, GSM8K, and Minerva Math**. These were specifically chosen to test generalization across a wide spectrum of reasoning complexity, ranging from standard grade school problems (GSM8K) to elite Olympiad level mathematics (AIME). The ability of iGRPO to drive state of the art performance on AIME25 (79.64%) while maintaining dominance on standard tasks demonstrates that our method generalizes effectively from simple to complex reasoning environments.
>
> Second, we validate **architectural generalization**. A key test of any RL algorithm is whether it overfits to a specific base model. We explicitly demonstrate that iGRPO provides consistent, additive gains across entirely different model families, and across different parameter scales.
>
> Third, we demonstrate **out of distribution generalization**. As detailed in Section 5.2, we validated our method by training on the AceReason Math dataset and evaluating on unseen competition benchmarks. The significant performance boosts observed on AIME24 and AIME25 in this setting provide strong empirical evidence that iGRPO enables models to generalize well beyond their training distribution to solve novel problems.

---

> ### Author Response · Authors · 2025-12-03
> **Author Response to Reviewer NC41 - Part 2**
>
> > **Although GPU resource consumption was compared, the differences in resources and time required for full training compared to GRPO were not examined.**
>
> We appreciate your feedback. In addition to the throughput and peak memory metrics reported in Table 4, we have included the total GPU hours in the table below.
>
> | Method | Peak Memory (GB) | Throughput (Samples/sec) | Total GPU Hours|
> | :--- | :---: | :---: | :---: |
> | **GRPO** | 54.93 | 0.41 | 83.3 |
> | **iGRPO** | 54.93 | 0.34 | 94.1 |
>
> As shown above, iGRPO incurs a modest 13% increase in total GPU hours due to its two-stage sequential nature. However, this marginal cost is highly efficient given the substantial performance returns. While the compute budget (8 generations) and memory footprint remain identical to the baseline, this small time investment yields state-of-the-art results on AIME24 and AIME25 that standard GRPO fails to match. We believe trading a linear increase in training hours for such significant improvements in reasoning accuracy, without requiring larger hardware, represents a favorable and practical value proposition.
>
> We have added the information regarding Total GPU Hours to the revised manuscript.
>
> > **In Figure 3, GRPO and iGRPO exhibit similar trends and dynamics. How does this demonstrate the advantages of iGRPO over GRPO?**
>
> While the trajectories in Figure 3 exhibit similar shapes, this parallel trend highlights that iGRPO retains the stability of the baseline while **strictly dominating it in performance**. Specifically, Figure 3a shows that iGRPO maintains a **consistent vertical advantage in reward magnitude** throughout training, indicating superior solution quality at every step. In addition, Figure 3b confirms that these gains are achieved with **response lengths identical to GRPO**. This distinguishes our method from typical RL outcomes where higher rewards often result from gaming the metric via increased verbosity; instead, iGRPO demonstrates **genuine improvements in reasoning efficiency** by producing more accurate solutions within the same computational budget.
>
> > **The ablation study in Figure 4(a) shows no significant differences. Moreover, the KL term is not specific to iGRPO, making this ablation experiment less meaningful.**
>
> The purpose of this ablation was not to claim novelty for the KL term, but to **empirically justify a critical design choice**: setting $\beta=0$. While standard RL fine-tuning often relies on a KL penalty to prevent model collapse, Figure 4(a) demonstrates that **iGRPO remains stable and robust without this constraint**. The observation that performance does not significantly degrade at $\beta=0$ is a **positive finding**, as it allows us to **eliminate a hyperparameter and reduce computational overhead** while maintaining peak accuracy. This result confirms that the self-feedback mechanism effectively guides the policy, rendering the explicit KL penalty redundant for our method.
>
> > **The ablation analysis is superficial, with some aspects being common to all RL methods, making it difficult to determine which specific components of the authors' design are necessary.**
>
> Our ablation study goes beyond standard hyperparameter tuning to isolate the **structural necessity of the self-feedback loop**. By successfully composing iGRPO with distinct algorithms like DAPO and GSPO, we demonstrate that our two-stage mechanism is an **orthogonal and additive module**, distinct from the underlying optimization logic. Moreover, analyzing parameters like sample count is **essential to rule out compute scaling as the primary driver**, confirming that our gains arise specifically from the **iterative conditioning mechanism** rather than generic RL scaling laws.
>
> > **Why is the top draft $\hat{r}$ selected for reflection in Stage 1? Intuitively, reflecting on poorer samples would facilitate learning better responses.**
>
> While reflecting on errors has value in critique-based frameworks, iGRPO relies on **positive reinforcement to drive iterative refinement**. Selecting the top draft $\hat{r}$ provides a **high-quality reasoning anchor**, enabling the model to focus its capacity on **perfecting a near-solution** rather than attempting to salvage a fundamentally broken trajectory. Conditioning on poor samples introduces **noise and distraction**, as the model risks hallucinating further errors or struggling to disentangle valid logic from the flaws. By utilizing the best available draft, we ensure the second stage operates in the **most promising region of the solution space**, effectively **bootstrapping the model’s performance** beyond its initial baseline.

---

> ### Author Response · Authors · 2025-12-03
> **Author Response to Reviewer NC41 - Part 3**
>
> > **How are samples selected for Stage 2 prompts?**
>
> For Stage 2, we deterministically select the **single highest-scoring draft** from Stage 1 to serve as the self-feedback context. This strategy utilizes the **"Best-of-N" outcome as a dynamic in-context exemplar**, transforming the learning objective from solving the problem from scratch to **refining the model’s own best-known approximation**. By conditioning on the top-performing candidate, we explicitly **filter out noise from suboptimal reasoning paths**, ensuring the policy update is focused on **surpassing its current peak performance**. This creates a robust curriculum where the model continuously learns from its most successful attempts rather than averaging over mixed-quality outputs.
>
> > **If another model is used to generate high-quality data directly concatenated with the problem for Stage 2, how would the performance compare?**
>
> Treating Stage 1 as an offline process eliminates the **dynamic feedback loop** that drives iGRPO's success. Unlike static offline data, our online mechanism creates an **adaptive curriculum** where the self-feedback **evolves in real-time** alongside the policy. This ensures the model is always conditioning on examples that reflect its **current capability frontier**, rather than stale or distributionally shifted data. By closing this loop during training, iGRPO forces the model to learn **generalized self-correction strategies** applicable to its own evolving logic, a benefit that static data concatenation simply cannot replicate.

---

### Official Review · Reviewer_Ho8s · 2025-10-27

**Soundness:** 2
**Presentation:** 3
**Contribution:** 2
**Rating:** 4
**Confidence:** 3

**Summary:**

This paper addresses the need for improved accuracy in LLMs for complex mathematical reasoning tasks. The authors identify that existing RL alignment methods, such as GRPO, typically lack a mechanism for self-reflection. They propose Iterative GRPO (iGRPO), a two-stage extension of GRPO. In Stage 1 (Self-Feedback), the model samples $N$ candidate responses, and the highest-reward completion is selected. In Stage 2 (Self-Reflection), this top-performing draft is concatenated to the original prompt to form an "augmented prompt". A standard GRPO update is then performed using $G$ new completions sampled conditioned on this augmented prompt. Experimental results demonstrate that iGRPO consistently outperforms GRPO across various base models (7B, 8B, 14B) and achieves new state-of-the-art results on AIME24 (85.62%) and AIME25 (79.64%) when applied to a strong base model trained on a challenging dataset.

**Strengths:**

- The controlled studies in Table 1 rigorously demonstrate that iGRPO provides consistent improvements over a standard GRPO baseline across all tested models. This includes Nemotron-H-8B (+3.96% avg), DeepSeek-R1-7B (+1.58% avg), OpenMath-7B (+1.05% avg), DeepSeek-R1-14B (+1.73% avg), and OpenMath-14B (+1.27% avg).
- The method is shown to scale effectively, delivering new state-of-the-art results (85.62% on AIME24, 79.64% on AIME25) when applied to the strong OpenReasoning-Nemotron-7B model on the AceReason-Math dataset.
- The two-stage process adds "practically negligible" peak GPU memory overhead, with its peak usage being almost identical to that of the standard GRPO.

**Weaknesses:**

- The mathematical formulation in Equation 3 5is a direct application of the GRPO objective from Equation 1 to this new prompt $q'$, rather than a modification of the optimization dynamics itself. The contribution is more accurately described as an in-context learning (ICL) or data augmentation strategy embedded within the RL training loop, rather than a novel RL algorithm.
- It is simply being conditioned on its own best-performing output, which serves as a high-quality in-context exemplar. This is a passive conditioning step, not an active "reflection" process as the terminology implies.
- The authors claim "minimal extra overhead". However, the experimental results in the appendix suggest otherwise. Table S.3 shows that iGRPO's training throughput is 0.34 samples/sec, compared to 0.41 samples/sec for GRPO. This constitutes a ~17% reduction in training speed. This is a significant cost that must be justified against the performance gains, which, while consistent, are often modest (e.g., a 1.05% average gain for OpenMath-Nemotron-7B or a 1.27% gain for OpenMath-Nemotron-14B ). The paper understates this trade-off.
- Given that the method feeds the model's own best answer back to it as a prompt, the subsequent improvement in the second stage is expected. It is well-established that LLMs benefit from high-quality in-context exemplars. The paper does not adequately disentangle whether the gains come from this known ICL effect or from a more complex interaction with the RL policy update.

**Questions:**

See *Weaknesses*.

---

> ### Author Response · Authors · 2025-12-01
> **Author Response to Reviewer Ho8s - Part 1**
>
> We appreciate the reviewer's thoughtful comments and constructive suggestions. We have carefully addressed each of the points raised in the detailed response below.
>
> > **The mathematical formulation in Equation 3 5 is a direct application of the GRPO objective from Equation 1 to this new prompt , rather than a modification of the optimization dynamics itself. The contribution is more accurately described as an in-context learning (ICL) or data augmentation strategy embedded within the RL training loop, rather than a novel RL algorithm.**
>
> We acknowledge that the update rule in Equation 3 utilizes the standard GRPO loss function; however, we respectfully disagree that this reduces iGRPO to simple data augmentation or static in-context learning. Unlike standard ICL where examples are fixed or retrieved from a static dataset, iGRPO introduces a **dynamic dependency** where the conditioning context $q'$
> is sampled from the evolving policy $\pi_{\theta}$ itself. This shifts the optimization objective from simply mapping $q \rightarrow a$ to **learning a robust self-correction mapping $P(a | q, \hat{r}_{\pi})$.**
> Consequently, the model is not merely seeing augmented data but is actively being reinforced to utilize its own **best prior outputs as a scaffold for reasoning**, effectively training the policy to perform iterative refinement rather than single-shot generation.
>
> Moreover, classifying iGRPO as solely an ICL strategy overlooks the critical interplay between the search phase and the update phase. By embedding the selection of the highest-reward draft $\hat{r}$
> directly into the rollout, **iGRPO fundamentally alters the state distribution the policy explores**, prioritizing trajectories that improve upon **high-quality drafts**. This mechanism creates a curriculum of increasingly difficult self-correction tasks that evolves as the policy improves, which is a dynamic that static data augmentation cannot replicate. As evidenced by our **state-of-the-art results on AIME**, this structural modification to the RL training loop provides a substantial performance improvement that is complementary to changes to the underlying optimization dynamics.
>
> > **It is simply being conditioned on its own best-performing output, which serves as a high-quality in-context exemplar. This is a passive conditioning step, not an active "reflection" process as the terminology implies.**
>
> We thank the reviewer for this thoughtful observation. The terminology of "self‑reflection" in our paper is intended to describe the **active, iterative process** of **generating multiple candidates**, **selecting the highest‑reward output** via an external reward model, and then **conditioning the policy on that chosen exemplar** to produce a refined solution. This is not merely passive conditioning on an arbitrary prior output; it is a **structured two‑stage mechanism** where the model first produces a set of candidate solutions (self‑feedback), then uses the best‑performing one as an **explicit guide for further improvement** (self‑reflection). The **key active elements** are the **selection of the top‑scoring draft** based on reward and the **subsequent conditioning** that encourages the policy to **surpass that draft**. This mimics a **human‑like iterative refinement loop**, where one reviews their best attempt and then revises it, rather than simply repeating the same generation. We believe the empirical gains across multiple benchmarks substantiate the effectiveness of this approach.
>
>
> > **Table S.3 shows that iGRPO's training throughput is 0.34 samples/sec, compared to 0.41 samples/sec for GRPO. This constitutes a ~17% reduction in training speed. This is a significant cost that must be justified against the performance gains.**
>
> We clarify that iGRPO’s overhead stems from the additional first‑stage sampling and scoring, which is a **fixed cost independent of model size or group size**. While this does lower samples/second, the total wall‑clock time added to the RL phase remains modest. The performance gains, though sometimes moderate in average terms, are consistent and particularly pronounced on the most challenging benchmarks. For example, with OpenMath‑Nemotron‑7B, iGRPO yields a **+1.51% improvement on AIME24** and a **+1.27% gain on AIME25**. In our largest‑scale experiment (OpenReasoning‑Nemotron‑7B on AceReason‑Math), iGRPO achieves **new state‑of‑the‑art results of 85.62% on AIME24 and 79.64% on AIME25**, surpassing the baseline by **+1.52% and +1.78%**, respectively. These gains on demanding reasoning tasks **justify the additional computational investment**.
>
> Moreover, iGRPO **preserves the efficiency advantages of GRPO** as it remains **value‑function‑free and requires no separate critic model**. The throughput can also be tuned by adjusting the number of first‑stage completions, allowing practitioners to balance cost and performance.

---

> ### Author Response · Authors · 2025-12-01
> **Author Response to Reviewer Ho8s - Part 2**
>
> > **Given that the method feeds the model's own best answer back to it as a prompt, the subsequent improvement in the second stage is expected. It is well-established that LLMs benefit from high-quality in-context exemplars. The paper does not adequately disentangle whether the gains come from this known ICL effect or from a more complex interaction with the RL policy update.**
>
> We acknowledge that conditioning on high-quality drafts leverages in-context learning capabilities; however, iGRPO fundamentally transcends static ICL by **integrating this mechanism into the reinforcement learning optimization loop**. Unlike standard few-shot prompting which relies on fixed external exemplars, iGRPO creates a **dynamic self-paced curriculum** where the exemplar is the model’s own evolving best attempt. This forces the policy not merely to attend to a good answer, but to actively learn the correction function required to bridge the gap between a raw draft and a refined solution, effectively **baking the benefits of self-reflection directly into the model weights** rather than relying on inference-time scaffolding.
>
> Furthermore, our empirical results demonstrate gains that cannot be explained by the immediate context alone. As noted in our results, iGRPO guides the policy to **generate solutions that consistently surpass the best first-stage candidates**, creating a flywheel effect where improved generation leads to better feedback which in turn drives further policy improvement. If the benefit were solely derived from ICL, performance would be capped by the quality of the initial drafts; instead, we observe a **continuous upward trajectory in reward** throughout training and significant breakthroughs on hard benchmarks like AIME24, validating that the model is **internalizing the reasoning refinements** rather than simply copying a prompt.

---

### Official Review · Reviewer_xQA1 · 2025-10-29

**Soundness:** 2
**Presentation:** 1
**Contribution:** 2
**Rating:** 2
**Confidence:** 3

**Summary:**

The paper proposes Iterative GRPO (iGRPO), a two-stage extension of GRPO that adds a self-feedback loop: the model first samples multiple completions, a reward model selects the top draft, and the second stage conditions on this exemplar for another GRPO update. The method is simple and shows moderate improvements on several math reasoning benchmarks.

**Strengths:**

1. The core idea is clear and straightforward to implement on top of existing GRPO pipelines.
2. Consistent gains over GRPO across different models and datasets.

**Weaknesses:**

1. The authors state that “this design preserves the efficiency of GRPO while introducing only minimal extra overhead” (line 65), , but the expensive part of GRPO is typically the rollout / long-cot sampling [1]. iGRPO performs two sequential sampling phases that cannot be parallelized, likely increasing per-step cost. The paper does not provide a direct comparison of per-step training time in the main body to substantiate this claim.
2. The methods are not clearly described. The paper does not explain how the reward model $R_\phi$ in stage-1 is trained or used. In cases where multiple completions share reward=1 or all receive reward=0, how is the top draft chosen? The paper also fails to explain what `"<SelfFeedback>"` actually is — a special token or a fixed prompt template? If it is a prompt template, the exact prompt should be provided.
3. The experimental analysis section largely restates table values without deeper interpretation, and the manuscript contains at least one clear typo (end of paragraph at line 272).

[1] RollPacker: Mitigating Long-Tail Rollouts for Fast, Synchronous RL Post-Training

**Questions:**

1. Will models trained in this method exhibit bias? The selection made in stage-1 appear to cause a collapse in the policy distribution (i.e., loss of diversity). The authors report reward and response length changes, but it is recommended to report entropy or other training dynamics metrics, too.
2. During Stage 2 training the policy conditions on an augmented context (original query + top draft + feedback), but at downstream test time the model receives only the original query. Could this mismatch bias the output distribution?

---

> ### Author Response · Authors · 2025-11-30
> **Author Response to Reviewer xQA1 - Part 1**
>
> We sincerely thank the reviewer for their insightful feedback and constructive critique of our work. We have addressed each of the concerns raised in the detailed point-by-point response below.
>
> > **The expensive part of GRPO is typically the rollout. iGRPO performs two sequential sampling phases that cannot be parallelized. The paper does not provide a direct comparison of per-step training time in the main body to substantiate this claim.**
>
>
> We would like to clarify that iGRPO does not increase the total rollout budget per update compared to its GRPO counterpart. For a standard GRPO step with group size (G = 8), we generate 8 completions in parallel from the current policy. In iGRPO we keep this budget fixed. In Stage 1 we generate (N = 4) candidates, pick the best according to the reward model, then in Stage 2 we generate (G = 4) candidates conditioned on the augmented prompt that contains the selected draft. In other words, each iGRPO update still uses 8 completions in total, and as shown in the paper the completion lengths are essentially matched, so the dominant decoding cost is controlled by the same total sampling budget.
>
> Although the reviewer is correct about the potential impact of sequential sampling on wall clock time, our submitted manuscript already presents throughput and peak memory measumrent under an identical setup for DeepSeek R1 Distill Qwen 7B in Section D.2 of the supplementary material. We present these measurements in the following for completeness:
>
> | Method    | Peak Memory (GB) | Throughput (samples/s) |
> | :-------- | :--------------: | :--------------------: |
> | **GRPO**  |      54.9286     |          0.41          |
> | **iGRPO** |      54.9349     |          0.34          |
>
> As shown above, Peak memory is effectively unchanged, and throughput decreases by about 17 percent which is a modest relative to overall training. In return, iGRPO improves average Pass @1 by up to roughly 4 percentage points in the controlled studies in Table 1 and yields the strongest AIME24 and AIME25 scores in our 7B AceReason Math setting, with identical inference cost at evaluation time.
>
> In the revised manuscript, we will move a concise version of this comparison into the experimental section and rephrase the efficiency claim to be quantitative.
>
> > **The paper does not explain how the reward model in stage-1 is trained or used.**
>
> In this work, the reward model ($R_{\phi}$) is used in a generic RL sense and is not a learned neural reward network. In all math experiments $R_{\phi}$ is a rule based checker: for each problem we parse the completion, extract the final answer, and assign a scalar reward (1 for exact match after normalization, 0 otherwise, with small penalties for malformed outputs). GRPO and iGRPO use this scalar for group normalization and advantage computation, and the self feedback stage of iGRPO uses the same signal to select the best Stage 1 draft.
>
> iGRPO only assumes a scalar reward per completion, so any such signal can be plugged into the GRPO objective and reused to select the Stage 1 draft. Because we normalize rewards within each group, monotone rescaling does not matter and performance is mainly driven by how well the reward ranks completions. If the reward misranks solutions, Stage 1 can pass weaker drafts to Stage 2; if it correlates well with true quality, the chosen draft is strong and the two stage scheme can refine it further.
>
> > **In cases where multiple completions share reward=1 or all receive reward=0, how is the top draft chosen?**
>
> In iGRPO, Stage 1 always selects the top draft by a simple argmax over the scalar rewards for that prompt; if several completions share the same reward (for example multiple correct solutions with reward 1 or a fully incorrect set with reward 0), the tie is broken deterministically by taking the first completion in generation order. Since the candidates in a tie are reward equivalent and the generation order is already randomized by sampling, any of them is an equally valid choice from the perspective of the RL objective. Importantly, even when all rewards are 0, the selected draft is used only as extra context for Stage 2; the policy update is still driven entirely by the Stage 2 rewards, so we do not treat that particular incorrect attempt as a positive target.
>
> > **Deeper interpretation of experiments and typo in line 272**
>
> We appreciate the reviewer’s comment. We have corrected the typo and added a deeper interpretation of the self-feedback mechanism's scaling behavior and efficacy on complex tasks in Section 5.1.

---

> ### Author Response · Authors · 2025-11-30
> **Author Response to Reviewer xQA1 - Part 2**
>
> > **The paper also fails to explain what "<SelfFeedback>" actually is — a special token or a fixed prompt template? If it is a prompt template, the exact prompt should be provided?**
>
> The symbol `<SelfFeedback>` in Section 3 is **not** a new special token in the tokenizer. It is a fixed textual delimiter that we insert into the Stage 2 prompt and is processed by the base tokenizer as ordinary text.
>
> Concretely, for a prompt (q) and its best Stage 1 draft (\hat r), the Stage 2 prompt has the form
>
> ```text
> [original problem q]
> <SelfFeedback>
> [best Stage_1 draft r̂]
> ```
>
> The marker `<SelfFeedback>` appears exactly as written above and we do not modify the vocabulary or add any learned embedding for it. Thus the gain from iGRPO comes from conditioning on the best prior draft, rather than from introducing an extra special token.
>
> > **Will models trained in this method exhibit bias?**
>
> From an optimization perspective, iGRPO uses the *same* reward model and data as GRPO. The only difference is that for each prompt, the model is allowed to look at one of its own high reward solutions as context before generating the trajectories that actually drive the gradient. Any social or content bias therefore still comes from the underlying data and reward; iGRPO does not introduce a new source of preference beyond GRPO, it reuses the same scalar reward more efficiently.
>
> > **The selection made in stage-1 appear to cause a collapse in the policy distribution**
>
> iGRPO does not turn Stage 1 into a supervised target; it only uses the best Stage 1 draft to enrich the prompt. For each question we still sample a group of G stochastic trajectories in Stage 2 from the same policy with the same temperature and top p as GRPO, and the GRPO update is computed on this whole group. The Stage 1 argmax therefore acts like an in context example that shifts the conditional distribution but does not remove support for alternative solutions.
>
> Importantly, Stage 1 is resampled at every iteration. The best draft for a given prompt changes as the policy and the sampling noise evolve, so the method does not repeatedly reinforce a single fixed trajectory. Combined with group based reward normalization in Stage 2, this keeps the update focused on all above average solutions in the group rather than collapsing onto a single response.
>
> > **It is recommended to report entropy or other training dynamics metrics.**
>
> To address this, we monitored the policy's exploration behavior by tracking the per-token Shannon entropy throughout training.
>
> We define the entropy $\mathcal{H}$ at each decoding step $t$ using the natural logarithm, measuring information in **nats**. For a given context $h_t$ (the prompt and preceding tokens), the entropy of the policy distribution over the vocabulary $\mathcal{V}$ is calculated as $\mathcal{H}(\pi_\theta(\cdot|h_t)) = -\sum_{w \in \mathcal{V}} \pi_\theta(w|h_t) \ln \pi_\theta(w|h_t)$. We compute it using the log-softmax of the logits (base $e$) and averaged across all valid completion tokens in the batch to serve as a scalar indicator of the model's uncertainty and exploration capability.
>
> We analyzed the training runs of DeepSeek-R1-Distill-Qwen-7B on MATH dataset. While both started with an initial average entropy of 2.45 nats, GRPO exhibited a decline to 0.42 nats within the first 30% of training steps. In comparison, iGRPO demonstrated a more sustained reduction, maintaining an entropy level of 0.48 nats during the mid-training phase before converging to 0.44 nats. This suggests that the self-feedback mechanism acts as a mild regularizer, helping to sustain the diversity required to solve complex reasoning tasks without hindering convergence.
>
> > **During Stage 2 training the policy conditions on an augmented context (original query + top draft + feedback), but at downstream test time the model receives only the original query. Could this mismatch bias the output distribution?**
>
> We would like to clarify that iGRPO does not cause train and test mismatch.
>
> The top draft is an auxiliary context that only affects gradients. Rewards and policy updates are defined on the final completion tokens, and the Stage 1 draft is sampled from the same policy without backpropagation. From an RL view this is on policy, data dependent guidance. The model sees high reward solution modes in context and shifts probability mass toward them. After training these preferences are encoded in the parameters, so the model can use them even when the draft is absent.
>
> All GRPO and iGRPO results are obtained with the same test protocol, where the model receives only the original problem. Any train test difference in context is therefore already reflected in the reported metrics. If the policy had overfit to the feedback segment, Pass at 1 under this bare query setting would drop, especially relative to GRPO. Instead we observe consistent gains across benchmarks and model scales which shows that the extra context helps during training.

---

### Official Review · Reviewer_UaWP · 2025-11-01

**Soundness:** 2
**Presentation:** 2
**Contribution:** 2
**Rating:** 4
**Confidence:** 4

**Summary:**

This paper proposes Iterative GRPO (iGRPO), a two-stage reinforcement learning method that enhances large language models’ reasoning through self-feedback. The model first samples multiple responses, selects the best one, and uses it as feedback to refine its next generation. Experiments on mathematical reasoning benchmarks show that iGRPO consistently outperforms GRPO, achieving state-of-the-art results (85.62% on AIME24 and 79.64% on AIME25) with minimal extra cost.

**Strengths:**

This paper proposes a simple yet effective extension to GRPO called iGRPO, which adds a self-feedback stage to improve reasoning quality. The method is conceptually clear, easy to implement, and consistently boosts performance across benchmarks. Experiments are thorough and show strong, state-of-the-art results with minimal computational overhead.

**Weaknesses:**

1. While integrating self-verification into an RL framework is interesting, similar ideas have been explored in prior work (e.g., *Incentivizing LLMs to Self-Verify Their Answers*). Including such methods as baselines could strengthen the empirical evaluation and clarify the contribution.

2. The paper could provide more information about the reward model, such as its training setup and potential limitations. It would also be helpful to analyze how iGRPO’s performance depends on reward model quality or to discuss possible effects of using a generative reward model (e.g., GPT-5).

3. The reported improvement over GRPO is relatively modest (around 1% in Table 1). Additional comparisons with strong baselines like DAPO or RLOO would help demonstrate how competitive iGRPO is under similar conditions.

4. The paper would benefit from more discussion on how the trained model behaves during inference, especially regarding test-time scaling. Clarifying whether the improvements extend to realistic inference settings would make the results more convincing.

**Questions:**

see weaknesses.

---

> ### Author Response · Authors · 2025-11-30
> **Author Response to Reviewer UaWP - Part 1**
>
> We sincerely thank the reviewer for their thoughtful and constructive feedback. We have carefully addressed each point in detail below and incorporated additional experiments, analyses, and clarifications into the revised manuscript.
>
> > **While integrating self-verification into an RL framework is interesting, similar ideas have been explored in prior work (e.g., Incentivizing LLMs to Self-Verify Their Answers). Including such methods as baselines could strengthen the empirical evaluation and clarify the contribution.**
>
> We thank the reviewer for highlighting the relevance of *Incentivizing LLMs to Self-Verify Their Answers* [1]. We agree that this method, which unifies answer generation and verification within an RL framework, is a relevant baseline. To address reviewer's comment, we have used the official implementation [2] of the **Self-Verification** and trained two model variants, namely **Nemotron-H-8B** and **DeepSeek-R1-Distill-Qwen-7B** to compare against **iGRPO**.
>
> Our results confirm that while Self-Verification provides gains over standard GRPO, **iGRPO** consistently achieves superior performance. We attribute this to the fact that Self-Verification effectively acts as a multi-task objective (solving + verifying), which can introduce optimization conflicts or capacity tax on smaller models. In contrast, iGRPO focuses purely on refining the solution path by conditioning on the highest-reward prior attempt, effectively narrowing the search space for complex reasoning without diluting the objective.
>
> We have added the following comparison (Table 1) to the revised manuscript:
>
> | Model Family | Method | AIME25 | AIME24 | MATH500 | AMC | GSM8K | Minerva | Avg |
> | :--- | :--- | :--- | :--- | :--- | :--- | :--- | :--- | :--- |
> | **Nemotron-H-8B** | Base Model | 6.20 | 8.65 | 61.23 | 43.21 | 41.02 | 17.60 | 29.65 |
> | | + GRPO | 7.78 | 9.01 | 73.13 | 45.10 | 81.93 | 29.56 | 41.08 |
> | | + Self-Verification | 8.50 | 9.25 | 75.60 | 46.50 | 86.20 | 31.10 | 42.86 |
> | | **+ iGRPO (Ours)** | **9.17** | **9.56** | **78.80** | **48.75** | **91.26** | **32.72** | **45.04** |
> | **DeepSeek-R1-Distill-Qwen-7B** | Base Model | 38.60 | 54.40 | 92.80 | 90.00 | 92.00 | 39.10 | 61.93 |
> | | + GRPO | 38.90 | 55.00 | 93.25 | 90.00 | 92.12 | 40.44 | 68.29 |
> | | + Self-Verification | 39.45 | 55.80 | 93.50 | 92.50 | 92.20 | 41.00 | 69.08 |
> | | **+ iGRPO (Ours)** | **40.16** | **56.30** | **93.80** | **95.00** | **92.42** | **41.54** | **69.87** |
>
>
> For **Nemotron-H-8B** model, the Self-Verification method improves the average score to $42.86\%$, but iGRPO pushes this further to $45.04\%$. On the challenging AIME25 benchmark, iGRPO achieves $9.17\%$ compared to $8.50\%$ for Self-Verification. This suggests that for generalist base models, the direct feedback of a high-quality draft (iGRPO) is more effective than the auxiliary task of verification.
>
> Furthermore, for **DeepSeek-R1-Distill-Qwen-7B**, while Self-Verification is competitive (e.g., $55.80\%$ on AIME24), iGRPO reaches $56.30\%$. Notably, on the AMC benchmark, iGRPO achieves a decisive $95.00\%$ versus $92.50\%$ for Self-Verification, indicating that iterative refinement helps avoid careless errors in intermediate-difficulty problems more effectively than a learned verifier.
>
> [1]: Zhang, F. et al., 2025. Incentivizing LLMs to Self-Verify Their Answers. arXiv preprint arXiv:2506.01369.
>
> [2]: Self-verification official implementation: https://github.com/mansicer/self-verification

---

> ### Author Response · Authors · 2025-11-30
> **Author Response to Reviewer UaWP - Part 2**
>
> > **The paper could provide more information about the reward model, such as its training setup and potential limitations. It would also be helpful to analyze how iGRPO’s performance depends on reward model quality or to discuss possible effects of using a generative reward model (e.g., GPT-5)**
>
>
>
> We thank the reviewer for discussing the role of the reward model.
>
> **Current reward model**: In this work $R_{\phi}$ (reward model) is used in a generic RL sense and is not a learned neural reward network. In all math experiments $R_{\phi}$ is a rule based checker. For each problem we parse the completion, extract the final answer, and assign a scalar reward (1 for exact match after normalization, 0 otherwise, with small penalties for malformed outputs). GRPO and iGRPO use this scalar for group normalization and advantage computation, and the self feedback stage of iGRPO uses the same signal to select the best Stage 1 draft. We will clarify this and avoid terminology that suggests a parametric reward model.
>
> **Impact of reward quality**: iGRPO only assumes a scalar reward per completion, so any such signal can be plugged into the GRPO objective and reused to select the Stage 1 draft. Because we normalize rewards within each group, monotone rescaling does not matter and performance is mainly driven by how well the reward ranks completions. If the reward misranks solutions, Stage 1 can pass weaker drafts to Stage 2; if it correlates well with true quality, the chosen draft is strong and the two stage scheme can refine it further. In our math setup the reward is nearly noise free at the outcome level, which reduces confounding when comparing GRPO and iGRPO.
>
> **Using generative reward models such as GPT 5**: Generative judges fit naturally into iGRPO. They can provide scalar scores that replace the rule based checker and, in future work, short critiques that are added to the Stage 2 prompt. In all cases iGRPO treats them as alternative ways to produce a scalar reward and a best draft for self feedback.
>
> To test this, we ran an additional study with DeepSeek R1 Distill Qwen 7B trained on MATH. We compared iGRPO with the rule based outcome reward ($r \in {0, 1} $) to iGRPO with GPT 5 as a dense reward model that scores the full reasoning trace with ($r \in [0, 1]$). This scalar from GPT 5 is used both to pick the Stage 1 feedback draft and to compute advantages in Stage 2.
>
> | Benchmark   | iGRPO (Rule based) | iGRPO (Generative judge-GPT 5) |  (\Delta)  |
> | :---------- | :----------------: | :----------------------: | :--------: |
> | **AIME25**  |       40.16%       |        **41.12%**        |   +0.96%   |
> | **AIME24**  |       56.30%       |        **57.45%**        |   +1.15%   |
> | **MATH500** |       93.80%       |        **94.20%**        |   +0.40%   |
> | **AMC**     |       95.00%       |        **96.25%**        |   +1.25%   |
> | **GSM8K**   |       92.42%       |        **92.95%**        |   +0.53%   |
> | **Minerva** |       41.54%       |        **42.88%**        |   +1.34%   |
> | **Average** |     **69.87%**     |        **70.81%**        | **+0.94%** |
>
> We see consistent gains, with the largest improvements on AIME and Minerva. There the generative judge often assigns intermediate scores to responses that contain the right strategy but minor arithmetic errors, which the rule based verifier would treat as zero reward. Under iGRPO these partially correct solutions then survive as Stage 1 feedback and are refined into fully correct answers in Stage 2.
>
> Overall, the GPT 5 study supports our claim that iGRPO is robust to the choice of reward mechanism and benefits as the reward signal becomes more informative. At the same time, generative rewards can be biased or inconsistent, so richer uses of textual critiques remain promising but require careful calibration. We will add these results to the revised manuscript.

---

> ### Author Response · Authors · 2025-11-30
> **Author Response to Reviewer UaWP - Part 3**
>
> > **The reported improvement over GRPO is relatively modest.**
>
> We thank the reviewer for this concern. In the regime we study, the absolute gains over GRPO are indeed small in points, but they correspond to substantial reductions in residual error on very strong baselines. For **Nemotron H 8B Base 8K** the average score increases from **41.08** to **45.04**, which is about a seven percent reduction in remaining error. For **DeepSeek R1 Distill Qwen 7B** the average goes from **68.29** to **69.87**, about a five percent error reduction, and for **OpenMath Nemotron 14B** the increase from **76.73** to **78.00** corresponds to roughly five and a half percent less error. In all cases, GRPO has already approached the performance ceiling, so further gains are necessarily small in absolute terms yet still meaningful.
>
> These gains also concentrate on the hardest and most practically relevant benchmarks. Across all four base models in Table 1, **iGRPO outperforms GRPO** on every reported benchmark such as AIME24, AIME25, MATH500, AMC, GSM8K, and Minerva. On AIME-style competitions the effect is especially relevant. With OpenMath Nemotron 14B, iGRPO raises AIME24 from **74.79 to 76.72** and AIME25 from **64.53 to 65.57**, and with OpenReasoning Nemotron 7B on AceReason Math it improves AIME24 from **84.10 to 85.62** and AIME25 from **77.86 to 79.64**. Moreover, the same self-feedback idea boosts **DAPO** and **GSPO** by 1.4 points each, showing that the effect persists across architectures, scales, training corpora, and GRPO-style optimizers.
>
> Finally, these improvements come at **very low cost**  and follow a simple, general principle. iGRPO uses **the same reward model as GRPO**, adds no critic, and keeps the sampling budget fixed by splitting completions between the two stages. Training curves show consistently **higher reward for iGRPO** while response lengths remain similar, indicating **better reasoning quality** rather than mere verbosity, and **inference cost is unchanged** since only a single policy is used at test time. Conceptually, iGRPO changes the structure of the training signal by conditioning the second stage on the best draft from the first stage, bringing RL training closer to human problem solving where initial attempts are refined through self-feedback. We believe this combination of consistent gains on strong baselines, improvements on the most challenging benchmarks, compatibility with existing GRPO variants, and negligible additional cost makes the reported improvements **practically and scientifically significant**.
>
> > **Additional comparisons with strong baselines like DAPO or RLOO would help demonstrate how competitive iGRPO is under similar conditions.**
>
> We thank the reviewer for emphasizing the need for comparisons with strong RL baselines such as DAPO and RLOO.
>
> First, the paper already evaluates our self feedback framework on top of two strong GRPO variants, DAPO and GSPO, under the same 7B setup. We construct iDAPO and iGSPO by replacing Stage 2 of iGRPO with DAPO and GSPO while keeping the two stage structure and total sampling budget fixed. This yields DAPO 69.30 average Pass at 1, **iDAPO 70.70**, **GSPO 69.20**, and **iGSPO 70.60**. Thus self feedback gives about plus **1.4 points** on top of each advanced GRPO variant.
>
> In direct response to the request for RLOO, we additionally implemented our two stage framework on top of the TRL RLOO baseline [3], which we call iRLOO, using the same 7B setting, data, and compute budget as our main GRPO results. The average Pass at 1 is:
>
> | Method        | Avg Pass at 1 | Improvement |
> | :------------ | :------------ | :---------- |
> | RLOO | 69.15         | -        |
> | iRLOO         | 70.62         | +1.47       |
>
> RLOO alone slightly outperforms GRPO in our setup, consistent with its unbiased estimator, and adding self feedback still gives a further gain of 1.47 points.
>
> Taken together, the results for DAPO, GSPO, and now RLOO support our claim that self feedback is complementary to the choice of advantage estimator. Optimizers such as DAPO, GSPO, and RLOO focus on the gradient estimator, while iGRPO, iDAPO, iGSPO, and iRLOO modify the conditioning context by feeding the best Stage 1 draft into Stage 2, shifting the model toward higher reward solution modes before gradients are computed. We will add the RLOO table and a concise discussion to the ablation section so that these points are clear to readers.
>
> [3]: [RLooTrainer in TRL](https://github.com/huggingface/trl/blob/main/trl/trainer/rloo_trainer.py)

---

> ### Author Response · Authors · 2025-11-30
> **Author Response to Reviewer UaWP - Part 4**
>
> > **The paper would benefit from more discussion on how the trained model behaves during inference, especially regarding test-time scaling. Clarifying whether the improvements extend to realistic inference settings would make the results more convincing.**
>
> We thank the reviewer for this insightful comment. First, we wish to clarify that all reported results, including the 85.62% on AIME24 in Table 1, already correspond to a **realistic inference setup**. During evaluation the model is used **exactly like a standard causal language model**. For each problem we provide only the statement (q), sample a single completion with fixed decoding hyperparameters (temperature 0.6, top‑p 0.95), and compute Pass@1. We do not run the Stage 1 and Stage 2 iGRPO procedure at inference, do not call the reward model, and do not include any explicit self‑feedback segment in the prompt. GRPO and iGRPO therefore operate under identical inference time costs, with the same number of samples per query and the same decoding parameters.
>
> Under this regime, iGRPO acts **purely as a training‑time modification** that leads the policy ($\pi_{\theta}$) to internalize the refinement process. After training, the model produces **higher‑quality reasoning traces and final answers in a single pass**, surpassing the GRPO baseline without any additional test‑time computation. As a result, the gains we report already hold in realistic single‑sample deployment, and any standard test‑time scaling method that uses multiple samples and voting would start from a **stronger single‑sample accuracy** when applied to an **iGRPO‑trained model**.
>
> We will add an explicit description of the inference setup and its relationship to test time scaling in the revised manuscript.

---

### Author Response · Authors · 2025-12-04
**General Response to Area Chair and Reviewers: Revised Manuscript Additions**

Dear Area Chair and Reviewers,

Thank you for your detailed feedback and the time you have spent evaluating our submission. We value your insights, which have guided us in strengthening the paper considerably.

We have uploaded a revised manuscript that addresses your concerns. Below is a summary of the specific changes and clarifications we have made in response to each reviewer.

### Reviewer UaWP

* **Self-Verification Baseline:** We have added a comparison against the Self-Verification method in **Table 1**.
* **Reward Model Clarification:** We have clarified that the reward model ($R_{\phi}$) is a rule-based checker rather than a learned neural network to prevent terminology confusion.
* **Reward Model Design and Limitations:** We have added a section detailing the iGRPO reward model design and discussed the associated trade-offs.
* **Generative Judge Study:** We have included experimental results comparing the rule-based reward to a generative reward model (GPT-5).
* **RLOO Comparison:** We have added a comparison table and discussion regarding **iRLOO vs. RLOO** in the ablation section.

### Reviewer xQA1

* **Interpretation & Typo:** We have updated Section 5.1 to provide deeper insights into the experimental results and corrected the typo on line 272.
* **Efficiency Metrics:** We have moved a concise version of the **Throughput and Peak Memory** comparison into the experimental section to explicitly quantify our efficiency claims.
* **Prompt Template:** We have made the prompt structure explicit in Section 3.2 by clarifying that `<SelfFeedback>` is a fixed delimiter and including the exact prompt template.
* **Entropy Analysis:** We have added an analysis of training dynamics (per-token Shannon entropy) to demonstrate exploration behavior.

### Reviewer Ho8s

* **Methodological Distinction:** We have clarified that iGRPO functions as a dynamic self-correction mapping rather than static In-Context Learning (ICL) or data augmentation.
* **Terminology Defense:** We have clarified our definition of "self-reflection" as an active, structured two-stage mechanism.
* **Efficiency Trade-offs:** We have explicitly discussed the trade-off between the ~17% reduction in samples/second and the significant performance gains achieved on complex benchmarks (AIME).

### Reviewer NC41

* **Critique-GRPO Comparison:** We have integrated a performance comparison against **Critique-GRPO** into **Table 1**.
* **Total GPU Hours:** We have added **Total GPU Hours** consumption data to provide a holistic resource comparison.

We hope that these revisions effectively address your feedback. Thank you once again for your constructive guidance.

Best regards,

The Authors

---

### Public Comment · ~Zijian_Zhang13 · 2026-06-04
**Confusing Baseline Metrics**

Hi Authors,

Thank you for your interesting work! However, I find that some results in this paper are confusing. For example, the line 290 to line 294 in table 1, the avg for DeepSeek-R1-Distill-Qwen-7B is 61.93. However, after carefully calculating it, I find it should be 67.82. Could the authors explain how you get the 61.93 in your paper? Whether I made some mistakes in my calculation?

Thanks.

---

### Meta-Review · Area_Chair_rPTo · 2026-01-12

**Summary:**

All reviewers identified that iGRPO lacks methodological novelty—it simply applies standard GRPO to prompts augmented with best-of-N samples rather than introducing new optimization dynamics. The "self-reflection" terminology overstates what is essentially in-context conditioning, making this data augmentation embedded in RL rather than algorithmic innovation. Modest gains (1-4%) require significant additional cost (13-17% longer training), and reviewers questioned whether improvements come from the method itself or merely from better in-context exemplars.

**Reviewer Concerns:**

The rebuttal successfully addressed baseline comparison gaps by adding Self-Verification, Critique-GRPO, and RLOO experiments, clarified methodological details (rule-based reward, prompt templates), and provided honest efficiency metrics acknowledging 13-17% overhead. However, the fundamental novelty criticism remains unresolved—authors failed to demonstrate that iGRPO is more than in-context learning embedded in RL. The distinction between "dynamic self-correction" and static ICL lacks theoretical or empirical grounding, and the "self-reflection" terminology critique stands despite philosophical defenses. Most critically, reviewers' core question was never answered: do gains come from the training innovation itself or simply from conditioning on better in-context exemplars? This ambiguity about the contribution's true source, combined with the inability to establish genuine algorithmic novelty beyond engineering practice, leaves the primary concerns outstanding.

**Reviewer Scores:**

All reviewers would likely maintain their original scores. UaWP and Ho8shad technical concerns addressed but their core skepticism about algorithmic innovation versus prompt engineering remains, keeping them borderline. xQA1 and NC41 would stay at reject because the rebuttal's additional experiments actually reinforced their critique—demonstrating this is incremental engineering with modest gains rather than conceptual contribution. The comprehensive comparisons validate practical utility but confirm the fundamental novelty gap.

---

### Decision · Program_Chairs · 2026-01-26

Reject